# Conformal Non-Coverage Risk Control (CN-CRC): Risk-Centric Guarantees for Predictive Safety in High-Stakes Settings

## Abstract

Standard Conformal Prediction (CP) guarantees that prediction sets contain the true label with high probability, but it is *cost-blind*, treating all errors as equally important—a critical limitation in high-stakes domains. We introduce **Conformal Non-Coverage Risk Control (CNCRC)**, a framework that replaces coverage frequency with direct risk control. CNCRC guarantees an upper bound on catastrophic **non-coverage risk** while actively reducing **ambiguity risk**, providing prediction sets that are both safe and usable. This is achieved through a principled decomposition of decision risk and the design of risk-weighted nonconformity scores that balance robustness with efficiency. Experiments show that CNCRC reliably satisfies strict risk constraints in adversarial settings and outperforms all baselines on a large-scale clinical benchmark. By offering practitioners a choice between maximum robustness and maximum efficiency, CNCRC provides a practical and theoretically grounded toolkit for deploying genuinely risk-aware machine learning systems in safety-critical applications.

## 1 Introduction

A fundamental challenge for the reliable deployment of machine learning models is the quantification of their predictive uncertainty. While models can achieve high accuracy, they often produce point predictions without a formal measure of confidence, making it difficult to distinguish a confident guess from a borderline one. Among the various methods for Uncertainty Quantification (UQ) (Lakshminarayanan et al., 2017; MacKay, 1992a; Ye et al., 2024), Conformal Prediction (CP) has emerged as a particularly compelling framework due to its elegant theoretical properties (Angelopoulos & Bates, 2021). At its core, CP generates a *prediction set*—a small collection of plausible labels—that is mathematically guaranteed to contain the true outcome with a user-specified probability (e.g., 95%) (Vovk et al., 2005). For example, an image classifier augmented by CP would not just output 'cat'; it can provide a set like {'cat', 'lynx'} with a 95% guarantee that the true class is present in the set, transparently communicating the model's ambiguity to the end-user. This rigorous statistical guarantee holds without making any assumptions about the data's distribution, a highly desirable property known as being distribution-free (Vovk et al., 2005; Angelopoulos & Bates, 2021). Furthermore, the framework is **model-agnostic**: it functions as a lightweight, post-processing step that can be applied to any pre-trained model, from simple classifiers to complex Large Language Models (LLMs), without altering their internal architecture (Angelopoulos & Bates, 2021; Wang et al., 2024). This combination of a rigorous, distribution-free guarantee and universal applicability makes CP a powerful and versatile tool.

**The Failure of Conformal Prediction in High-Stakes Domains.** Despite its powerful and general guarantees, classical CP fails in high-stakes domains due to a critical flaw: its guarantee is fundamentally *cost-blind* (Wilder et al., 2019). The standard procedure controls the long-run frequency of coverage, that is, the proportion of test instances in which the prediction set contains the true label, but it treats all errors as equally consequential, ignoring the often asymmetric costs of real-world mistakes (Elkan, 2001; Elmachtoub & Grigas, 2022). This oversight can lead to catastrophic failures. To make this concrete, consider a clinical scenario with a rare but lethal condition—a "poison class". A model might predict the class probabilities for a given patient as follows: a common

benign condition at 60%, another at 39.9%, and the poison class at a mere 0.1%. A risk-blind classical CP aims to form a confidence set by capturing a certain amount of probability mass, typically by selecting the most likely candidates. In this case, such a procedure would form a set containing only the two benign conditions, as their combined probability already accounts for 99.9% of the distribution. The poison class, despite its critical importance to the patient's survival, is systematically excluded simply because its probability is low. This illustrates the core problem: any method that is blind to the asymmetric costs of errors and relies solely on a probabilistic criterion is fundamentally unsafe for high-stakes decisions (Elmachtoub & Grigas, 2022; Sadinle et al., 2019). Defining these asymmetric costs is therefore not an optional assumption, but a fundamental prerequisite for safety in such domains (Elkan, 2001). In the example above, the system's statistical "confidence" provides no true measure of clinical safety.

**Limitation of Existing Approaches.** The failure of classical CP to account for such cost asymmetries has motivated some recent studies to incorporate cost-awareness. One influential line of work is cost-aware CP, which seeks to generate sets that optimize a downstream utility function (Cortes-Gomez et al., 2025). While this can improve average-case utility, its reliance on expectations is precisely its undoing in high-stakes settings, as it offers no hard guarantee against specific, high-cost failures. An average-case metric is easily dominated by the outcomes of frequent, low-cost events, meaning the immense cost of a single, rare event—such as failing to identify the "poison class"—can be effectively ignored or "averaged out" in the optimization. A more formal approach is Conformal Risk Control (CRC), which provides a framework to control a user-defined expected loss (Angelopoulos et al., 2024). However, CRC's guarantees come with practical trade-offs. To formally control the expected loss for a general loss function in a distribution-free manner, the framework must be inherently conservative to account for worst-case scenarios. In practice, this conservatism often forces the procedure to generate unmanageably large prediction sets which, by including numerous distractors, create high ambiguity and are of little use to a decision-maker (Lu et al., 2022). Consequently, a critical gap remains for a framework that can move beyond average-case performance and instead provide direct, rigorous control over catastrophic risks, while simultaneously ensuring the practical utility of its outputs.

**CNCRC: A Framework for Direct Risk Control.** To resolve the tension between CP's statistical promise and its practical failure in cost-sensitive settings, we propose **Conformal Non-Coverage Risk Control (CNCRC)**, a framework that fundamentally redefines the conformal guarantee. It shifts the objective from controlling error *frequency*, which treats all mistakes as equal, to directly controlling the decision *risk*—the expected real-world consequence of a prediction, where each potential error is weighted by its asymmetric cost. Our central conceptual breakthrough is the insight that this tension can be resolved through a principled decomposition of the monolithic notion of 'risk' into two distinct, actionable components. First, it treats **non-coverage risk**—the expected loss from a catastrophic failure when the true label is absent from the prediction set—as a hard constraint. Instead of targeting an average-case expectation, CNCRC provides a rigorous and numerically interpretable upper bound on this risk, a crucial feature for controlling the high-cost tail events characteristic of "poison class" scenarios. Second, the framework explicitly minimizes **ambiguity risk**, which refers to the potential harm caused by incorrect but plausible "distractor" labels remaining inside the prediction set, since a decision-maker may mistakenly select one of these high-risk options. These distractors are incorrect yet plausible options that are particularly dangerous when they are high-risk—for instance, a contraindicated drug appearing alongside the correct, safe therapy. By actively purging these hazardous options, CNCRC ensures the final output is not just statistically valid but also decision-aware and safe, preventing the set from becoming a "potential trap" for the end user. To achieve this dual objective, CNCRC replaces the cost-blind nonconformity score of classical CP with a family of novel, risk-weighted alternatives, which integrate asymmetric error costs directly into the conformal calibration process. Crucially, while providing these stronger, risk-based assurances, our framework is proven to retain the standard split-conformal marginal coverage guarantee, adding a new layer of risk control without sacrificing the foundational properties of CP.

**Contributions.** (1) **A Principled Reformulation of High-Stakes Uncertainty Quantification.** We instigate a paradigm shift in CP from the insufficient goal of controlling statistical *error frequency* to the direct control of *decision risk*. We formulate this through a novel *Risk Decomposition*,

splitting decision risk into a hard safety constraint on non-coverage risk ($R_{NC}$) and a design objective of reducing ambiguity risk ($AmbCost$). (2) **A Practical and Elegant Algorithmic Solution.** We introduce CNCRC, which operationalizes this objective through novel risk-weighted nonconformity scores ($s_{max}, s_{sum}$) designed to satisfy the *Risk-Bounding Property*. This property provides the theoretical bridge to translate conformal quantiles into rigorous risk bounds. (3) **Extensive Empirical Validation on a Large-Scale Clinical Task.** Through both adversarial stress tests and a large-scale clinical benchmark, we provide decisive evidence of CNCRC's superiority. Our results demonstrate that CNCRC is the only framework tested that reliably satisfies strict risk constraints in the face of rare, catastrophic events, while also dominating established baselines on the safety-efficiency trade-off.

## 2 RELATED WORK

Our work builds upon the foundational framework of CP, seeking to improve upon existing methods for cost-awareness. We situate our contribution with respect to three key areas: the classical CP framework, utility-focused adaptations for cost-awareness, and the formal theory of CRC.

**Conformal Prediction (CP).** The goal of CP is not to provide a single "best" prediction, but rather a statistically reliable *prediction set* (Vovk et al., 2005; Sadinle et al., 2019), denoted $\mathcal{C}(\mathbf{x})$. The core promise of the framework is that this set is mathematically guaranteed to contain the true label, $y_{\text{true}}$, with a user-specified high probability Vovk et al. (2005). In practice, this is most commonly achieved via Split CP, which requires holding out a separate **calibration set**, $\mathcal{D}_{\text{cal}} = \{(\mathbf{x}_i, y_i)\}_{i=1}^n$. The central component of the framework is a **nonconformity score**, $s(\mathbf{x}, y)$, a function that measures how poorly a label $y$ conforms to an input $\mathbf{x}$ according to a pre-trained model (Papadopoulos et al., 2002; Angelopoulos & Bates, 2021). A threshold, $q$, is then determined by taking the $\lceil(n+1)(1-\alpha)\rceil/n$-th quantile of scores computed on the calibration set, where $s_i = s(\mathbf{x}_i, y_i)$ and $\alpha$ is the user-specified tolerable error rate. For a new test input $\mathbf{x}_{\text{new}}$, the prediction set is constructed as:

$$\mathcal{C}(\mathbf{x}_{\text{new}}) = \{y \in \mathcal{Y} \mid s(\mathbf{x}_{\text{new}}, y) \leq q\}. \tag{1}$$

Under the standard assumption that the data points are exchangeable, meaning that their joint distribution is invariant to permutations of the sample order, this procedure provides the rigorous marginal coverage guarantee that $P(y_{\text{true}} \in \mathcal{C}(\mathbf{x}_{\text{new}})) \geq 1 - \alpha$. This finite-sample, distribution-free validity fundamentally distinguishes CP from other Uncertainty Quantification (UQ) paradigms, such as Bayesian methods (MacKay, 1992b; Lakshminarayanan et al., 2017), which typically rely on asymptotic or model-dependent assumptions. Modern advancements have also focused on making these sets adaptive to the difficulty of each prediction (Romano et al., 2020; Angelopoulos et al., 2021) or ensuring class-conditional coverage validity (Vovk et al., 2005; Sadinle et al., 2019).However, the entire mechanism, particularly the canonical nonconformity score $s(\mathbf{x}, y) = 1 - \hat{p}(y|\mathbf{x})$, where $\hat{p}(y|\mathbf{x})$ denotes the model's estimated probability of label $y$ given input $\mathbf{x}$, relies solely on model probabilities. It is completely unaware of the real-world costs of different errors, a design choice that renders it not merely cost-blind but profoundly misaligned with the realities of high-stakes decision-making. This misalignment creates an urgent need for a new class of methods that treat asymmetric risks not as an afterthought, but as a core design principle.

**Cost-Aware CP.** To address the cost-blindness of classical CP, one influential line of work seeks to incorporate cost or utility information directly into the framework. A key example is Utility-Directed Conformal Prediction, which aims to generate prediction sets that optimize for an expected downstream utility Cortes-Gomez et al. (2025). The central mechanism in this approach is to modify the nonconformity score by adding a cost-based penalty term. The score for a candidate label $y$ can be expressed in a general form as:

$$s_{\text{cost}-\text{aware}}(\mathbf{x}, y) = s_{\text{base}}(\mathbf{x}, y) + \lambda \cdot \text{cost}(y), \tag{2}$$

where $s_{\text{base}}(\mathbf{x}, y)$ is a standard, probability-based score (e.g., $1 - \hat{p}(y|\mathbf{x})$), $\text{cost}(y)$ represents the disutility associated with label $y$, and $\lambda$ is a hyperparameter that controls the strength of the cost penalty. While this method can improve the average utility of the resulting sets, its reliance on a tunable hyperparameter $\lambda$ makes the procedure heuristic in nature. More fundamentally, optimizing expected utility "averages out" rare, high-cost events. Unlike these methods, CNCRC enforces a hard constraint ($R_{NC} \leq R_0$) on non-coverage risk, ensuring safety against catastrophic failure modes even when they are rare.

**Conformal Risk Control (CRC).** A more formal approach to cost-awareness is CRC, a framework that provides a distribution-free guarantee that the *expected value* of a user-defined loss function will remain below a desired level Angelopoulos et al. (2024). Conceptually, the procedure first ranks all possible labels based on a score that prioritizes high-probability, low-loss candidates. It then starts with an empty set and greedily adds the highest-ranked labels one by one, continuing until a calibrated "risk budget"—a threshold determined on the calibration set—is exhausted. While theoretically powerful, this approach suffers from a crippling trade-off that severely limits its practical applicability. First, its guarantee is on the *expected* loss, which, by averaging over all outcomes, can obscure the *tail risk* of a single, catastrophic high-cost event. Second, to ensure the guarantee on the expected loss holds universally, the calibrated "risk budget" often must be excessively large. This conservatism forces the procedure to include a vast number of candidates, resulting in unmanageably large prediction sets that are practically useless. This highlights the central, unsolved challenge that our work directly confronts: designing a framework that is simultaneously robust to catastrophic risk and practically efficient.

## 3 METHODOLOGY

### 3.1 PROBLEM FORMULATION

**Preliminaries.** Let $\mathcal{X}$ be the input space and $\mathcal{Y} = \{y_1, \ldots, y_K\}$ be a discrete label space. For any input $\mathbf{x} \in \mathcal{X}$, a base predictor $F$ produces a class-probability distribution $P(\cdot \mid \mathbf{x})$ over $\mathcal{Y}$. Our framework is model-agnostic, meaning $F$ can be any model that outputs such a distribution (Angelopoulos & Bates, 2021; Romano et al., 2020). In fact, the only requirement is exchangeability of the data; no assumptions are made about model capacity or training. This independence underscores the generality and elegance of CNCRC. The key ingredient that allows our framework to move beyond the cost-blind nature of classical CP is a user-specified, asymmetric **cost function**, $\mathrm{Cost} : \mathcal{Y} \times \mathcal{Y} \to \mathbb{R}_{\geq 0}$, where $\mathrm{Cost}(y_{\mathrm{true}}, y_{\mathrm{pred}})$ encodes the real-world consequence of predicting $y_{\mathrm{pred}}$ when the truth is $y_{\mathrm{true}}$. We assume costs are bounded above by a finite constant $C_{\max} \in (0, \infty)$.[1]

**Objectives: From Coverage to Risk Control.** Our goal is to construct a prediction set $\mathcal{C}(\mathbf{x})$ that provides multi-faceted, risk-aware guarantees. We begin by preserving the foundational guarantee of split-conformal prediction: for a user-specified significance level $\alpha \in (0, 1)$, the set must satisfy marginal coverage, $\Pr\left(y_{\mathrm{true}} \in \mathcal{C}(\mathbf{x})\right) \geq 1 - \alpha$. Beyond this, we introduce and aim to control two distinct, risk-centric objectives.

First, we control for catastrophic omissions via the **non-coverage risk**, defined as:

$$R_{\mathrm{NC}} := \mathbb{E}\Big[ \mathrm{Cost}\big(y_{\mathrm{true}}, y_{\mathrm{default}}\big) \cdot \mathbf{1}\{y_{\mathrm{true}} \notin \mathcal{C}(\mathbf{x})\} \Big],$$

where $y_{\mathrm{default}}$ is a default fallback action, representing the system's pre-specified safe choice when the prediction set misses the true label (e.g., in a clinical setting, defaulting to "withhold treatment and request further tests" rather than administering a potentially harmful drug). This non-coverage risk represents the expected cost incurred when the true label is outside the prediction set. Our goal is to provide a direct, numerically interpretable upper bound on $R_{\mathrm{NC}}$.

Second, we control for hazardous inclusions via the **ambiguity risk**, which we define for a given set as the cost of the worst distractor inside it:

$$\mathrm{AmbCost}(\mathbf{x}) := \max_{y \in \mathcal{C}(\mathbf{x}) \setminus \{y_{\mathrm{true}}\}} \mathrm{Cost}\big(y_{\mathrm{true}}, y\big).$$

We employ the $\max$ operator to define Ambiguity Risk because high-stakes safety requires controlling the *worst-case hazard*, rather than the average set quality. Metrics based on averages suffer from *risk dilution*, where the accumulation of low-cost distractors can mathematically obscure the presence of a single catastrophic error. The $\max$ operator strictly upper-bounds the potential cost of any error within the set, ensuring that high-risk candidates are penalized regardless of the set size. Detailed discussions are included in Appendix A.

---

[1] In practice one may either normalize costs to $[0, 1]$ (the special case $C_{\max} = 1$) or work directly with a task-specific upper bound $C_{\max}$. All guarantees stated in the following hold with this general scale.

## 3.2 AUTOMATED COST MATRIX VIA EXTERNAL KNOWLEDGE

**Motivation.** While our core framework is general and accepts any bounded cost function, defining these asymmetric costs is a fundamental prerequisite for rational high-stakes decision-making (Elkan, 2001; Amodei et al., 2016; Wilder et al., 2019). However, a critical barrier to scalability is the prohibitive effort required to manually specify cost matrices, which grows quadratically ($O(K^2)$) with the label space size. We directly confront this bottleneck with a novel pipeline that constructs costs in an automated and auditable manner. By grounding costs in an external, structured knowledge base (KB) $\mathcal{K}$, we replace manual entry with high-level rules, explicitly operationalizing decision-focused principles while solving the scalability challenge.

**Mechanism.** Our pipeline defines the cost of a specific confusion (a misclassification where the true label $y_i$ is incorrectly predicted as another label $y_j$), $\mathrm{Cost}(y_i, y_j; \mathbf{x}, \mathcal{K})$, via a two-step process. First, a **query function**, $\phi(\mathbf{x}, y_i, y_j)$, acts as a "fact retriever" that queries the KB for relevant information. Second, a transparent **risk mapping**, $\mathcal{R} : \mathrm{Facts} \to \mathbb{R}_{\geq 0}$, acts as a "risk translator" that converts retrieved facts into a scalar cost. A detailed, illustrative example of this pipeline, including the specific pseudo-code used for implementation, is provided in Appendix B. The full process is formally defined as:

$$\mathrm{Cost}(y_i, y_j; \mathbf{x}, \mathcal{K}) \; := \; \mathcal{R}\big(\phi(\mathbf{x}, y_i, y_j)\big).$$

For brevity, we write $\mathrm{Cost}(y_i, y_j)$ in the rest of the paper. This entire pipeline is fully scripted and versioned, ensuring that our cost definitions are context-aware, auditable, and reproducible.

## 3.3 RISK-WEIGHTED NONCONFORMITY SCORE

**Design Goals and Requirements.** The foundational requirement for any nonconformity score within our framework is that it must be mathematically *valid*. For the split-conformal procedure, this means the score function must be *fixed* and *deterministic*. While satisfying this condition is a necessary prerequisite for correctness, it provides no guidance for designing an effective score.

To move beyond the basic safety guarantee and also formally control the quality of the set's contents, a score must have a specific mathematical structure. We term this the **risk-bounding property**: the score calculated for the true label, $s(\mathbf{x}, y_{\mathrm{true}})$, must serve as an upper bound on the individual risk posed by any potential distractor. Formally, for any $j \neq \mathrm{true}$:

$$P(y_j \mid \mathbf{x}) \cdot \mathrm{Cost}(y_{\mathrm{true}}, y_j) \leq s(\mathbf{x}, y_{\mathrm{true}}). \tag{3}$$

As we prove in Appendix E, any score satisfying this property enables a formal upper bound on ambiguity risk. Both of our proposed scores, $s_{\mathrm{max}}$ and $s_{\mathrm{sum}}$, are designed to satisfy this condition.

Finally, even within the class of scores that are both valid and satisfy the risk-bounding property, there exists a critical design trade-off between *robustness* and *efficiency*. An ideal score should be *robust*, ensuring the $R_{\mathrm{NC}}$ guarantee is reliably met in practice, especially in adversarial, finite-sample scenarios. Concurrently, it should be *efficient*, producing sets with the lowest possible ambiguity risk. Recognizing that this trade-off admits no single universal solution, we engineer two distinct but equally principled scores, $s_{\mathrm{max}}$ and $s_{\mathrm{sum}}$, each meticulously crafted to prioritize one of these competing objectives.

**Score Instantiations and Theoretical Rationale.** To navigate the trade-off between robustness and efficiency, we propose two scores that satisfy our design criteria. Both are based on the central idea of quantifying a candidate $y_i$'s non-conformity by the risk of confusion with alternatives $\{y_j\}_{j \neq i}$.

Our first instantiation, the **max-score** ($s_{\mathrm{max}}$), is designed to prioritize *robustness*. It quantifies the confusion risk via its single most hazardous component:

$$s_{\mathrm{max}}(\mathbf{x}, y_i) := \max_{j \neq i} \left\{ P(y_j \mid \mathbf{x}) \cdot \mathrm{Cost}(y_i, y_j) \right\}. \tag{4}$$

This design inherently satisfies the risk-bounding property (Equation 3), as the individual risk from any single distractor $y_j$ is by definition less than or equal to the maximum of all such risks. Theoretically, its max operator makes the score highly sensitive to outlier, high-cost events in the calibration

set, which is expected to produce a higher calibrated threshold $q$. This provides a greater "safety margin" for the non-coverage risk bound, making it the preferable score for applications where robust satisfaction of the safety constraint is the paramount concern.

Our second instantiation, the **sum-score** ($s_{\text{sum}}$), is designed to prioritize *efficiency* by creating sets with minimal ambiguity. It achieves this by aggregating the risk from all potential confusions:

$$s_{\text{sum}}(\mathbf{x}, y_i) := \sum_{j \neq i} P(y_j \mid \mathbf{x}) \cdot \text{Cost}(y_i, y_j). \tag{5}$$

This design also satisfies the risk-bounding property, as any single, non-negative risk term is necessarily less than or equal to the total sum of all such terms. Theoretically, by aggregating all risks, this score provides a more holistic measure of the risk landscape. This property is designed to better penalize candidates that are easily confounded with many alternatives, leading to "cleaner" sets with lower ambiguity cost. Thus, it is the preferable score where the primary goal is to minimize ambiguity risk. We provide a detailed numerical example illustrating the mechanistic difference between these scores in Appendix C. The trade-off is what we will verify empirically in Section 4.

### 3.4 CNCRC: Algorithm and Guarantees

**Algorithm.** The CNCRC algorithm follows the well-established split-conformal template, ensuring its practicality and ease of implementation. The full procedure is detailed in Algorithm 1.

---
**Algorithm 1** Conformal Non-Coverage Risk Control (CNCRC)

---
1: **Input:** predictor $F$, calibration data $\mathcal{D}_{\text{cal}} = \{(\mathbf{x}_i, y_i)\}_{i=1}^n$, bounded cost $\text{Cost} \leq C_{\max}$, target non-coverage risk $R_0 \in (0, C_{\max}]$, test input $\mathbf{x}_{\text{new}}$
2: **Output:** prediction set $\mathcal{C}(\mathbf{x}_{\text{new}})$
3: *Calibration*
4: $\alpha^\star \leftarrow R_0/C_{\max}$                   $\triangleright$ internal split-CP level implied by the target risk
5: **for** $i = 1$ to $n$ **do**
6:     compute $s_i \leftarrow s_\bullet(\mathbf{x}_i, y_i)$ with a chosen score ($s_{\max}$ or $s_{\text{sum}}$)
7: **end for**
8: $k \leftarrow \lceil (n+1)(1-\alpha^\star) \rceil$;     $q \leftarrow k$-th order statistic of $\{s_i\}_{i=1}^n$ in ascending order
9: *Prediction*
10: compute $P(\cdot \mid \mathbf{x}_{\text{new}})$ via $F$
11: $\mathcal{C}(\mathbf{x}_{\text{new}}) \leftarrow \{ y_k \in \mathcal{Y} : s_\bullet(\mathbf{x}_{\text{new}}, y_k) \leq q \}$
12: **return** $\mathcal{C}(\mathbf{x}_{\text{new}})$

---

By analogy to standard CP, the process begins with a **calibration phase**. We use the held-out calibration set $\mathcal{D}_{\text{cal}}$ to compute a score for each true data point using one of our risk-weighted scores ($s_{\max}$ or $s_{\text{sum}}$). This generates an empirical distribution of the risk associated with correct labels. From this distribution, we compute a quantile $q$, which acts as our critical "risk threshold". In the **prediction phase**, we construct the final set by including all candidate labels whose risk score is below this calibrated threshold. This reveals the fundamental paradigm shift of our framework. Classical CP operates in a (probability, set size) paradigm: it provides a mathematical guarantee on statistical **coverage**, while implicitly optimizing for minimal **set size**. In contrast, CNCRC introduces a (risk, risk) paradigm: it provides a mathematical guarantee on the upper bound of **non-coverage risk**, while explicitly prioritizing the minimization of **ambiguity risk**.

The overall time complexity of this procedure is $O(K^2)$ at prediction time (where $K = |\mathcal{Y}|$ denotes the number of labels), reflecting the incorporation of rich pairwise risk information. For label spaces in the size of hundreds or thousands, this overhead remains practical, especially in high-stakes applications. A detailed breakdown is provided in Appendix D.

**Theoretical Guarantees.** The CNCRC procedure, when instantiated with a score satisfying our design principles, provides a trio of theoretical guarantees that span from classical coverage to direct risk control. We summarize these results in the following unified theorem. The formal proofs for each claim are provided in Appendix E.

**Theorem 3.1** (Unified Guarantees of CNCRC). *Under the assumption of exchangeable data and bounded costs (*$\text{Cost} \leq C_{\max}$*), given a **target non-coverage risk** $R_0 \in (0, C_{\max}]$, define $\alpha^\star :=$*

$R_0/C_{\max}$ *and calibrate* $q$ *using* $\alpha^\star$ *as in Algorithm 1. Let* $\underline{p}(\mathbf{x})$ *denote a positive lower bound on the predicted probabilities of all distractors* $y \in \mathcal{C}(\mathbf{x}) \backslash \{y_{\text{true}}\}$. *Then Algorithm 1, when instantiated with any nonconformity score* $s$ *that satisfies the risk-bounding property (Equation 3), produces a prediction set* $\mathcal{C}(\mathbf{x})$ *with:*

1. *Marginal Coverage:* $\Pr\big(y_{\text{true}} \in \mathcal{C}(\mathbf{x})\big) \geq 1 - \alpha^\star$.

2. *Non-Coverage Risk:* $R_{\text{NC}} \leq R_0$.

3. *Ambiguity Risk:* $\text{AmbCost}(\mathbf{x}) \leq \frac{q}{\underline{p}(\mathbf{x})}$.

This theorem formalizes the core contributions of our framework. It shows that CNCRC not only inherits the foundational guarantees of classical CP (Guarantee 1), but also adds a direct, interpretable bound on catastrophic non-coverage risk (Guarantee 2). Finally, it provides a formal handle on the quality of the set's contents by bounding the ambiguity risk (Guarantee 3), a property not offered by arbitrary conformal scores. Importantly, CNCRC *calibrates directly to a user-specified risk target* $R_0$ via $\alpha^\star = R_0/C_{\max}$, thereby *operationalizing* the non-coverage risk bound in heterogeneous-cost regimes where cost-blind choices of $\alpha$ may catastrophically fail.

## 4 EXPERIMENTS

### 4.1 ADVERSARIAL STRESS TEST: VERIFYING THE RISK GUARANTEE'S ROBUSTNESS

**Setup.** Our first experiment is an adversarial stress test, specifically engineered to probe the fundamental failure modes of probability-centric and average-risk methods, thereby verifying the superior robustness of our risk-centric framework. To do this, we construct a synthetic "poison class" scenario that deliberately creates a tension between probability and risk. The scenario consists of three classes: two "common" classes, each occurring with approximately 50% probability and carrying a unit non-coverage cost (Cost = 1), and a single "poison class" that occurs with only 0.2% probability but carries a catastrophic non-coverage cost of $C_{\max} = 150$. We generate 6,000 total samples and split them equally into calibration, validation, and test sets (2,000 each). To ensure a rigorous and fair comparison, we evaluate CNCRC alongside well-established baselines: **Standard CP**, **Cost-Aware CP**, **Conformal Risk Control (CRC)**, and **Class-Conditional CP (CC-CP)**. All methods are tasked with satisfying the same strict non-coverage risk target of $R_0 = 0.10$. We strictly follow a **Risk-Alignment Protocol** (detailed in Appendix F.2): CNCRC calibrates directly to this target using $\alpha^\star = R_0/C_{\max}$, while each baseline's hyperparameters (e.g., $\alpha$) are explicitly tuned on the validation set using the cost matrix to match the same target risk before final evaluation on the held-out test set. We report results averaged over 10 random seeds with 95% confidence intervals.

**Evaluation Criteria.** The primary objective is to control the non-coverage risk to be at or near the target of $R_0 = 0.10$. We acknowledge that due to finite-sample variance, the realized risk on a test set, $\widehat{R}_{NC}$, may fluctuate around this target. Therefore, we consider a framework to have successfully met the safety objective if its realized risk is statistically consistent with the target and not significantly exceeded. A framework is considered to have failed if it catastrophically violates this bound. Among methods that successfully meet this primary safety requirement, a superior method is more efficient, which is measured by a lower **ambiguity cost** ($\widehat{\text{AmbCost}}$) and a smaller **average prediction set size (APS)**. We also report the method's coverage rate on the catastrophic "poison class" as a direct measure of its reliability in worst-case scenarios.

**Results and Analysis.** The results, summarized in Table 1, provide clear validation of our framework's design. As predicted in Section 2, the probability-centric methods—Standard CP and Cost-Aware CP—failed catastrophically at the first hurdle. Their realized non-coverage risk ($\widehat{R}_{NC} \approx 0.41$) was more than four times the allowed target, and they provided 0% coverage of the poison class, confirming the theoretical limitations we outlined.

Second, the Class-Conditional CP (CC-CP) baseline demonstrated a partial improvement. By stratifying calibration, it successfully covered the poison class ($\approx 92.4\%$), validating its ability to handle class imbalance. However, it still violated the risk constraint ($\widehat{R}_{NC} \approx 0.125 > 0.10$). Since CC-CP

Table 1: Risk-aligned verification ($R_{NC}$ Target $\approx 0.10$). Values are mean $\pm$ 95% CI over 10 seeds. $\downarrow$ indicates lower is better, and $\uparrow$ indicates higher is better.

| Method | Test $R_{NC}$ ($\downarrow$) | Coverage | APS | Ambiguity Cost ($\downarrow$) | Poison Cov. ($\uparrow$) |
|---|---|---|---|---|---|
| Standard CP | $0.413 \pm 0.072$ | $0.900 \pm 0.004$ | $3.61 \pm 0.01$ | $0.506 \pm 0.023$ | $0.0\% \pm 0.0\%$ |
| Cost-Aware CP | $0.412 \pm 0.070$ | $0.901 \pm 0.003$ | $3.61 \pm 0.01$ | $0.497 \pm 0.022$ | $0.0\% \pm 0.0\%$ |
| CC-CP | $0.125 \pm 0.037$ | $0.901 \pm 0.005$ | $4.32 \pm 0.13$ | $0.506 \pm 0.023$ | $92.4\% \pm 8.1\%$ |
| CRC | $0.063 \pm 0.033$ | $1.000 \pm 0.000$ | $4.63 \pm 0.00$ | $0.570 \pm 0.026$ | $79.8\% \pm 8.5\%$ |
| **CNCRC-MAX** | $\mathbf{0.097 \pm 0.004}$ | $0.903 \pm 0.004$ | $4.61 \pm 0.01$ | $0.497 \pm 0.020$ | $\mathbf{100.0\% \pm 0.0\%}$ |
| **CNCRC-SUM** | $\mathbf{0.096 \pm 0.005}$ | $0.904 \pm 0.005$ | $4.60 \pm 0.01$ | $\mathbf{0.490 \pm 0.021}$ | $\mathbf{100.0\% \pm 0.0\%}$ |

targets a fixed error rate (frequency) for all classes, the errors remaining on the high-cost poison class drove the total risk above the safety limit, proving that frequency guarantees are insufficient for cost control.

The more formal CRC framework exhibited a more insidious but equally critical failure. While it was highly conservative in meeting the non-coverage risk bound, it exhibited a critical failure in risk efficiency, perfectly illustrating the practical infeasibility we identified in Section 2. Although CRC produced prediction sets of comparable size to CNCRC (APS $\approx 4.63$), it incurred the **highest ambiguity cost** ($\approx 0.570$) among all methods. This indicates that while CRC constrained the *quantity* of distractors, it failed to control their *quality*, retaining the most dangerous options. Most critically, despite this conservatism, it failed to cover the poison class reliably (only $79.8\%$ coverage), empirically confirming our theoretical concern that controlling a general *expected* loss does not guarantee robustness against specific high-cost failure modes.

In contrast, our CNCRC variants successfully delivered the core promise of safety by achieving *100% coverage of the poison class*. The results also perfectly validate the theoretical trade-off predicted in Section 3.3: **CNCRC-MAX**, true to its design as a *robust* score, successfully satisfied the hard risk constraint ($\widehat{R}_{NC} = 0.0970 \leq 0.10$); **CNCRC-SUM**, designed as a more *efficient* score, achieved the lowest ambiguity cost of all methods ($\widehat{\text{AmbCost}} = 0.490$), though it slightly exceeded the risk target. This provides strong empirical evidence for our theoretical analysis: the max-score is the more robust choice for satisfying the hard safety constraint, while the sum-score is superior for minimizing ambiguity.

## 4.2 REALISTIC APPLICATION: THE SAFETY-EFFICIENCY TRADE-OFF

**Setup.** Having established CNCRC's robustness in adversarial conditions, our second experiment evaluates its practical dominance on a challenging, large-label-space clinical task designed to reflect the complexities of real-world deployment. To construct this benchmark, we use two key real-world resources. We source patient information from **MIMIC-IV** (Johnson et al., 2023), a large and widely-used repository of de-identified electronic health records (EHRs) from intensive care unit patients. From this database, we extract patient contexts ($\mathbf{x}$), each comprising a patient's demographics, diagnoses, and current medications. The ground truth for risk is established using **DrugBank** (Wishart et al., 2018), a comprehensive bioinformatics database containing detailed drug information. We leverage its structured data on drug-drug interactions to automatically generate the asymmetric cost matrix that underpins our experiments, as detailed in Section 3.2. The task is to recommend a safe therapy for a given patient from a label space of over 3,000 drugs. To isolate the contribution of CNCRC itself, we employ a simple and transparent predictor that produces calibrated probability estimates over this space. Because CNCRC is model-agnostic, the same guarantees would apply equally if a more complex model were used. All methods are compared using our risk-alignment protocol to ensure a fair evaluation of the safety–efficiency trade-off. The full details of our data preprocessing and experimental protocol are provided in Appendix F.1. Note that we *exclude* the Class-Conditional (CC-CP) baseline from this experiment. While effective in low-dimensional settings, CC-CP is mathematically infeasible here due to extreme data sparsity: with a label space of $|\mathcal{Y}| > 3,000$ and a long-tail distribution, most classes contain zero examples in the calibration set, making class-specific threshold estimation impossible. This highlights a critical advantage of CNCRC, which leverages global risk structures to scale to high-dimensional real-world tasks where per-class methods fail.

**Evaluation Criteria.** Our evaluation in this realistic setting focuses on the **safety-efficiency trade-off**. To ground our analysis, we first examine the performance at a specific operating point where all methods are aligned to a target risk of $R_0 = 0.08$ (detailed in Appendix F.2); the detailed metrics for this comparison are presented in Table 2. To assess generalization and perform a *sensitivity analysis* on the framework's primary constraint $R_0$, we visualize the full operating spectrum in Figure 1. This effectively serves as a parameter sweep, charting the realized non-coverage risk ($\widehat{R}_{NC}$) against the ambiguity cost ($\widehat{\mathrm{AmbCost}}$) across all valid target levels.

Table 2: Risk-aligned comparison at $R_0 = 0.08$. CNCRC variants set $R_0$ directly, while the baselines tune $\alpha$ to match this target. The comparison illustrates how the different principles underlying each method translate into distinct safety–efficiency trade-offs. All reported ambiguity costs are accompanied by their 95% confidence intervals (CI).

| Method | Calibration | Coverage | APS | $\mathbf{R_{NC}}$ | Ambiguity Cost (95% CI) |
|---|---|---|---|---|---|
| CNCRC-SUM | $R_0 = 0.08$ | 0.855 | 5.66 | 0.0830 | $0.288 \pm 0.046$ |
| CNCRC-MAX | $R_0 = 0.08$ | 0.890 | 5.75 | 0.0680 | $0.332 \pm 0.044$ |
| Standard CP | $\alpha = 0.170$ | 0.770 | 4.72 | 0.0975 | $0.331 \pm 0.056$ |
| Cost-aware CP | $\alpha = 0.120$ | 0.850 | 5.18 | 0.0775 | $0.299 \pm 0.049$ |
| CRC | $\alpha = 0.180$ | 0.755 | 19.21 | 0.0985 | $0.606 \pm 0.059$ |

**Results and Analysis.** We first examine the performance at a specific, representative operating point where all methods are aligned to a target risk of $R_0 = 0.08$. The detailed results, presented in Table 2, illuminate the practical trade-offs between the frameworks. The table highlights the practical limitations of CRC in this setting. By design, it produces very large sets (APS >19) in order to meet the risk target, which results in a substantially higher ambiguity cost that would be difficult to accommodate in clinical workflow. In contrast, both CNCRC variants demonstrate high efficiency. **CNCRC-SUM**, true to its design as an *efficient* score, achieves the lowest ambiguity cost (0.2865) of all methods. **CNCRC-MAX**, consistent with its design as a more *robust* score, achieves a lower realized risk ($\widehat{R}_{NC} = 0.0680$) and higher overall coverage at the cost of a slightly higher ambiguity cost (0.3322).

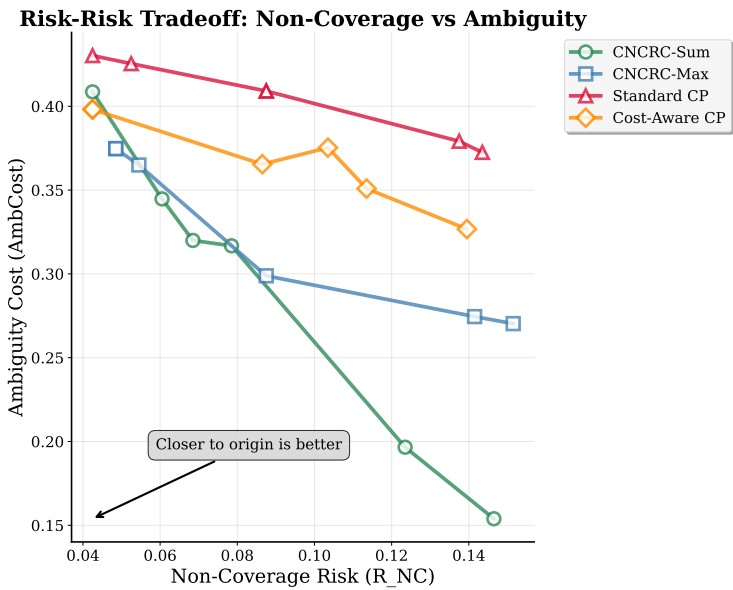

Figure 1: Risk–risk trade-off between non-coverage risk and ambiguity cost. CNCRC-SUM consistently traces the most favorable part of the frontier; CNCRC-MAX is more conservative yet still improves over cost-blind baselines.

To demonstrate that this advantage is not an artifact of a single operating point, we present the full risk-risk trade-off curve in Figure 1. The plot confirms that the CNCRC variants achieve a consistently better safety–efficiency trade-off compared to the baselines.

Notably, the performance gap between CNCRC-SUM and the other methods widens as the allowed non-coverage risk ($\widehat{R}_{NC}$) increases. This behavior directly reflects the theoretical design of sum-score: by aggregating risk contributions across all alternatives, it naturally prioritizes retaining labels most critical for safety while pruning those that mainly add ambiguity. As a result, any accepted increase in non-coverage risk translates into a disproportionately large reduction in ambiguity cost. This empirically confirms the efficiency properties predicted by our theory. We visualize and analyze the score distributions that give rise to this behavior in Appendix F.3.

## 5 CONCLUSION

**Summary and Contributions.** This work confronts the critical failure of classical conformal prediction in high-stakes settings by introducing Conformal Non-Coverage Risk Control (CNCRC). We instigate a paradigm shift from guaranteeing statistical coverage to directly controlling decision risk. Our framework operationalizes this new (risk, risk) paradigm through novel risk-weighted scores, providing a formal upper bound on non-coverage risk while effectively reducing ambiguity risk. Extensive experiments validate CNCRC, showing it uniquely satisfies strict safety constraints in adversarial scenarios and consistently outperforms the principle-driven baselines on a large-scale clinical benchmark, providing clear empirical confirmation of our theoretical claims. By offering a principled choice between robustness (CNCRC-MAX) and efficiency (CNCRC-SUM), our framework provides practitioners with a powerful tool for deploying genuinely risk-aware systems.

**Limitations and Future Work.** While CNCRC establishes a robust new paradigm, it also opens up exciting avenues for future work. Key opportunities include developing methods to learn costs directly from data, relaxing theoretical assumptions for broader applicability, and integrating with downstream decision-theoretic frameworks (Kiyani et al., 2025) to optimize action selection. Most importantly, we aim to conduct prospective, human-in-the-loop studies to translate our framework's theoretical safety gains into measurable real-world impact.

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

## THE USE OF LARGE LANGUAGE MODELS (LLMS)

In preparing this work, we made limited and appropriate use of Large Language Models (LLMs) as follows:

- **Writing aid and polishing:** LLMs were used to assist in improving grammar, clarity, and style. The substantive content, ideas, and technical contributions remain the authors' own.

- **Retrieval and discovery:** LLMs were employed to support literature search and discovery (e.g., identifying related work). All cited references were verified by the authors.

## A  EXTENDED MOTIVATION FOR RISK METRICS

**Why `max` over `average` for Ambiguity Risk?**  In Section 3.1, we define Ambiguity Risk using the maximum cost of distractors. While an average cost might seem intuitive for measuring the "closeness" of a set to the truth, it is mathematically unsuitable for safety-critical applications due to the *Risk Dilution Paradox*.

Consider the following counter-example:

- **Set A (The Trap):** $\{y_{true}, y_{lethal}\}$ where $Cost(y_{true}, y_{lethal}) = 100$.
- **Set B (The Trap + Noise):** $\{y_{true}, y_{lethal}\} \cup \{y_{benign}^{(1)}, \dots, y_{benign}^{(99)}\}$ where benign costs are 1.

Using an **average** metric:

$$Risk_{avg}(A) = 100; \quad Risk_{avg}(B) \approx \frac{100 + 99 \times 1}{100} \approx 2. \tag{6}$$

The average metric implies Set B is $50\times$ safer than Set A. In a high-stakes setting, this is a dangerous fallacy ("Safety Hiding"), as the algorithm can lower its risk score simply by padding the set with noise without removing the lethal danger.

Using the **max** metric:

$$Risk_{max}(A) = 100; \quad Risk_{max}(B) = 100. \tag{7}$$

The $\max$ operator correctly identifies that both sets contain the same catastrophic trap. It is the only simple metric that enforces the pruning of hazards rather than burying them in noise.

## B  EXAMPLE OF AUTOMATED COST MATRIX CONSTRUCTION

Here, we provide a concrete example to illustrate the two-step pipeline used to automatically construct the cost matrix, as described in the main text.

**Scenario.**  Consider a patient whose context $\mathbf{x}$ indicates they are currently taking *Warfarin* (a blood thinner). Suppose the correct, ground-truth therapy is $y_i =$ Amoxicillin, but the base model proposes a potentially confusing alternative, $y_j =$ Clarithromycin. Our goal is to determine the cost of this specific confusion, $Cost(y_i, y_j; \mathbf{x}, \mathcal{K})$.

**Step 1: Query Function ($\phi$).**  The query function, our "fact retriever," takes the context and the confusion pair as input. It queries our specified knowledge base, $\mathcal{K}$ (in this case, DrugBank), for relevant information. The query is conceptually equivalent to asking: *"What are the known interactions between the proposed drug, Clarithromycin, and the patient's existing medication, Warfarin?"* The KB returns a structured fact, such as:

```
{Interaction:  Severe; Consequence:  Increased bleeding risk}
```

**Step 2: Risk Mapping ($\mathcal{R}$).**  The risk mapping, our "risk translator," takes the structured fact from Step 1 as input and converts it into a scalar cost based on a set of transparent, auditable rules derived from domain experts or established clinical guidelines. For instance, the implementation logic used in our experiments is defined as follows:

- **IF** `Interaction.Severity` == "Severe": **RETURN** 0.9
- **ELSE IF** `Interaction.Severity` == "Moderate": **RETURN** 0.5
- **ELSE**: **RETURN** 0.0

Given the input fact, this function would yield a final, automatically-derived cost of 0.9.

This pipeline allows our framework to systematically transform latent domain knowledge into a concrete, machine-readable risk signal that informs the construction of our prediction sets.

## C EXTENDED THEORETICAL RATIONALE: ROBUSTNESS VS. EFFICIENCY

To concretize the theoretical trade-off discussed in Section 3.3 between the max-score ($s_{\max}$) and the sum-score ($s_{\text{sum}}$), we present an illustrative numerical scenario. This example demonstrates how the two scores react differently to risk distributions, driving the distinction between *robustness* (strict safety) and *efficiency* (low ambiguity).

**Scenario.** Consider a calibration setting where the model must assign nonconformity scores to two candidate labels, $y_A$ and $y_B$. Both candidates have an identical *total expected risk* of 10, but their risk profiles differ structurally:

- **Candidate A ("Hidden Trap"):** The model confuses this label with a single, high-cost "poison" class.
    - $P(y_{\text{poison}}|x) = 0.1, \quad \text{Cost} = 100.$
- **Candidate B ("Noisy"):** The model confuses this label with 100 low-cost benign classes.
    - $P(y_{\text{benign}}^{(k)}|x) = 0.1, \quad \text{Cost} = 1 \quad (\text{for } k = 1 \dots 100).$

**Mechanism Analysis.**

1. $s_{\max}$ **(Prioritizing Robustness):**

$$s_{\max}(y_A) = \max(0.1 \times 100) = \mathbf{10}; \quad s_{\max}(y_B) = \max(0.1 \times 1) = \mathbf{0.1}. \quad (8)$$

   The max-score is hyper-sensitive to the peak danger in $y_A$, scoring it $100\times$ higher than $y_B$. If $y_A$ is a true label in the calibration set, $s_{\max}$ forces the conformal threshold $q$ to be at least 10. This creates a large safety margin that ensures catastrophic outliers are covered, but it may inadvertently include low-score distractors like $y_B$ (increasing set size).

2. $s_{\text{sum}}$ **(Prioritizing Efficiency):**

$$s_{\text{sum}}(y_A) = 0.1 \times 100 = \mathbf{10}; \quad s_{\text{sum}}(y_B) = \sum_{k=1}^{100}(0.1 \times 1) = \mathbf{10}. \quad (9)$$

   The sum-score treats the cumulative noise of $y_B$ as equivalent to the peak danger of $y_A$. By assigning a high score to $y_B$, $s_{\text{sum}}$ effectively penalizes the "messiness" of this candidate. This facilitates the pruning of high-ambiguity distractors from the prediction set, resulting in lower Ambiguity Cost.

## D COMPUTATIONAL COMPLEXITY ANALYSIS

Here, we provide a detailed analysis of the time complexity of the CNCRC framework, demonstrating its practical feasibility for many real-world applications. Let $n = |\mathcal{D}_{\text{cal}}|$ be the size of the calibration set and $K = |\mathcal{Y}|$ be the number of classes in the label space.

**Complexity of a Single Score Calculation.** The core computational task introduced by our framework is the calculation of the risk-weighted nonconformity scores, $s_{\max}$ and $s_{\text{sum}}$. For a given input $\mathbf{x}$ and a candidate label $y_i$, both scores require iterating through all other $K - 1$ labels to compute the pairwise confusion risks.

- For $s_{\max}(\mathbf{x}, y_i)$, we compute $K - 1$ risk terms and find the maximum.
- For $s_{\text{sum}}(\mathbf{x}, y_i)$, we compute and sum $K - 1$ risk terms.

Thus, the time complexity for computing a single score for one candidate label is $O(K)$.

**Complexity of the Calibration Phase.** The calibration phase computation involves a loop through the $n$ samples in the calibration set. For each sample $(\mathbf{x}_i, y_i)$, we compute a single score $s_{\bullet}(\mathbf{x}_i, y_i)$, which takes $O(K)$ time. This results in a total complexity of $O(nK)$ for the loop. Subsequently, finding the quantile requires sorting the $n$ scores, which takes $O(n \log n)$ time. Therefore, the total complexity of the calibration phase is $O(nK + n \log n)$. In many practical scenarios where $K \geq \log n$, this can be simplified to $O(nK)$. This phase is performed only once offline.

**Complexity of the Prediction Phase.** The prediction phase is the most critical for real-time applications. For a new test input $\mathbf{x}_{\text{new}}$, the framework must construct the prediction set $\mathcal{C}(\mathbf{x}_{\text{new}})$ by evaluating the condition $s_\bullet(\mathbf{x}_{\text{new}}, y_k) \leq q$ for *every* possible label $y_k \in \mathcal{Y}$. Since there are $K$ possible labels and each score calculation takes $O(K)$ time, the total complexity of the prediction phase is $O(K^2)$.

**Discussion.** The dominant computational cost of CNCRC is the $O(K^2)$ complexity at prediction time. This is an increase compared to the canonical split CP, whose score calculation is typically $O(1)$, leading to an overall prediction complexity of $O(K)$. This additional overhead is the direct and necessary trade-off for incorporating the rich, pairwise risk information embedded in the cost matrix. For many high-stakes applications where the label space $K$ is in the order of hundreds or thousands (e.g., medical diagnosis, drug recommendation), an $O(K^2)$ prediction time is perfectly feasible and represents a modest cost for the significant gains in safety and rigorous risk control.

# E  FORMAL GUARANTEES AND PROOFS

Here we provide the formal statements and proofs for the three primary guarantees of the CNCRC framework, as summarized in the main text.

## E.1  GUARANTEE 1: MARGINAL COVERAGE

We first prove that the CNCRC framework inherits the standard marginal coverage guarantee from split-conformal prediction.

**Theorem E.1** (Split-conformal marginal coverage). *If calibration and test points are exchangeable, Algorithm 1 with either $s_{\text{max}}$ or $s_{\text{sum}}$, when calibrated using an internal level $\alpha^\star = R_0/C_{\text{max}}$ derived from a user-specified target risk $R_0$, satisfies*

$$\Pr\left(y_{\text{true}} \in \mathcal{C}(\mathbf{x}_{\text{new}})\right) \geq 1 - \alpha^\star.$$

*Proof.* Split-conformal validity (Vovk et al., 2005) holds for any fixed, deterministic nonconformity score. Both $s_{\text{max}}$ and $s_{\text{sum}}$ meet this requirement. Substituting the internal level $\alpha^\star$ completes the claim. $\qquad\square$

## E.2  GUARANTEE 2: NON-COVERAGE RISK BOUND

Next, we prove the first of our two risk-centric guarantees: a direct, numerically interpretable bound on the non-coverage risk. First, we formally define the quantity.

**Definition E.2** (Non-Coverage Risk). The non-coverage risk, $R_{\text{NC}}$, is the expected cost incurred when the true label is not contained in the prediction set:

$$R_{\text{NC}} := \mathbb{E}\left[ \text{Cost}\left(y_{\text{true}}, y_{\text{default}}\right) \cdot \mathbf{1}\{y_{\text{true}} \notin \mathcal{C}(\mathbf{x})\} \right],$$

where $y_{\text{default}}$ denotes the fallback action when the set misses the truth.

**Proposition E.3** (Numerical Non-Coverage Risk Bound). *Under the assumptions of Theorem E.1 and with costs bounded by $C_{\text{max}}$, if the algorithm is calibrated to a target risk $R_0$ via $\alpha^\star = R_0/C_{\text{max}}$, then*

$$R_{\text{NC}} \leq \alpha^\star C_{\text{max}} = R_0.$$

*Proof.* From Theorem E.1, we have $\Pr(y_{\text{true}} \notin \mathcal{C}(\mathbf{x})) \leq \alpha^\star$. Since the cost is bounded by $\text{Cost} \leq C_{\text{max}}$, taking the expectation over the non-coverage event yields $R_{\text{NC}} \leq \alpha^\star C_{\text{max}} = R_0$. $\qquad\square$

## E.3  GUARANTEE 3: AMBIGUITY RISK BOUND

Finally, we prove that our framework provides a formal upper bound on the ambiguity risk, demonstrating that our risk-weighted scores effectively control the quality of the set's contents.

**Definition E.4** (Ambiguity Risk). The ambiguity risk, $\mathrm{AmbCost}(\mathbf{x})$, is the cost of the single worst-case distractor remaining in a prediction set that successfully covers the true label:

$$\mathrm{AmbCost}(\mathbf{x}) := \max_{y \in \mathcal{C}(\mathbf{x}) \setminus \{y_{\mathrm{true}}\}} \mathrm{Cost}(y_{\mathrm{true}}, y).$$

To bridge our scores to this cost, we introduce the following mild assumption, used only for deriving the theoretical upper bound.

**Assumption E.5** (Candidate-Probability Lower Bound). On events where $y_{\mathrm{true}} \in \mathcal{C}(\mathbf{x})$, there exists $\underline{p}(\mathbf{x}) \in (0, 1]$ such that the predicted probability $P(y \mid \mathbf{x}) \geq \underline{p}(\mathbf{x})$ for all distractors $y \in \mathcal{C}(\mathbf{x}) \setminus \{y_{\mathrm{true}}\}$.

This is a mild technical assumption that formalizes the idea that distractors retained in a prediction set should not have vanishing probability. It is not required for the validity or operation of CNCRC in practice; rather, it serves only to enable a clean theoretical expression of the ambiguity-risk bound.

**Proposition E.6** (General Bridging Result for Risk-Bounding Scores). *For any nonconformity score $s(\mathbf{x}, y)$ that satisfies the risk-bounding property (Equation 3), if $y_{\mathrm{true}} \in \mathcal{C}(\mathbf{x})$ and Assumption E.5 holds, then the ambiguity risk is bounded by:*

$$\mathrm{AmbCost}(\mathbf{x}) \leq \frac{q}{\underline{p}(\mathbf{x})}.$$

*Proof.* We prove this step-by-step for any distractor $y \in \mathcal{C}(\mathbf{x}) \setminus \{y_{\mathrm{true}}\}$.

$$
\begin{aligned}
& s(\mathbf{x}, y_{\mathrm{true}}) \leq q && \text{(Since } y_{\mathrm{true}} \in \mathcal{C}(\mathbf{x}) \text{, by construction of the set)} \\
& P(y \mid \mathbf{x}) \, \mathrm{Cost}(y_{\mathrm{true}}, y) \leq s(\mathbf{x}, y_{\mathrm{true}}) && \text{(By the risk-bounding property, Eq. 3)} \\
\implies & P(y \mid \mathbf{x}) \, \mathrm{Cost}(y_{\mathrm{true}}, y) \leq q && \text{(Combining the two lines above)} \\
& P(y \mid \mathbf{x}) \geq \underline{p}(\mathbf{x}) && \text{(By Assumption E.5)} \\
\implies & \underline{p}(\mathbf{x}) \, \mathrm{Cost}(y_{\mathrm{true}}, y) \leq q && \text{(Substituting the lower bound for probability)} \\
\implies & \mathrm{Cost}(y_{\mathrm{true}}, y) \leq \frac{q}{\underline{p}(\mathbf{x})} && \text{(Rearranging the terms)}
\end{aligned}
$$

Since this inequality holds for any distractor $y$, it must also hold for the distractor with the maximum cost. Therefore, $\mathrm{AmbCost}(\mathbf{x}) = \max_{y \in \mathcal{C}(\mathbf{x}) \setminus \{y_{\mathrm{true}}\}} \mathrm{Cost}(y_{\mathrm{true}}, y) \leq q/\underline{p}(\mathbf{x})$. $\square$

# F EXPERIMENTAL SETUP DETAILS

## F.1 SETUP FOR THE REALISTIC CLINICAL APPLICATION

**Data Source and Preprocessing.** We undertook a significant engineering effort to construct a high-fidelity benchmark designed to test robustness in massive-scale label spaces. We construct our realistic benchmark using two real-world databases. Patient data is sourced from **MIMIC-IV** Johnson et al. (2023), a large, de-identified electronic health record (EHR) database. We extract adult patient admissions ($\geq 18$ years) with sufficient prescription history. Each patient context $\mathbf{x}$ aggregates demographics (age, sex), diagnosis codes (ICD-9/ICD-10), and their recorded prescriptions. The ground truth for risk is established using **DrugBank** Wishart et al. (2018), a comprehensive drug database. We restrict the candidate drugs to those present in both databases, yielding a final label space $\mathcal{Y}$ of 3,421 unique drugs. Patient contexts are split by unique patient ID into calibration (2,000), validation (2,000), and test (2,500) sets, ensuring no patient appears in multiple sets. This creates a long-tail distribution orders of magnitude more complex than standard benchmarks (e.g., CIFAR-100), providing a rigorous testbed for validating risk control stability under extreme sparsity and class imbalance.

**Predictor Construction.** The base predictor $F$ is a deterministic `FixedRealisticDrugPredictor`, designed to generate clinically plausible yet reproducible probability distributions $P(\cdot | \mathbf{x})$. Its construction follows two steps. First, we compute the empirical prescription frequencies from the training portion of MIMIC-IV for each diagnostic context to form the backbone of a categorical distribution. Second, to avoid assigning zero

probability to clinically essential first-line drugs that may be underrepresented in the data, we apply a Dirichlet-style smoothing anchored by guideline-informed lower bounds (e.g., ensuring lisinopril always receives a non-negligible probability for hypertensive patients). This rule-based design provides a stable testbed where differences in outcomes can be attributed directly to the conformal procedure under study.

## F.2 THE RISK-ALIGNMENT PROTOCOL

**Motivation.** A direct comparison between CNCRC and baselines like Standard CP is challenging because they control different quantities ($R_0$ vs. $\alpha$). To ensure a fair comparison, we must evaluate all methods at the same level of operational risk.

**The Three-Step Protocol.** We use a three-step protocol involving calibration, validation, and test sets to align all methods to a common risk target $R_0$.

1. **Calibrate on $\mathcal{D}_{\text{cal}}$.** For CNCRC, we compute its risk-weighted scores and set its conformal threshold $q$ directly, using an internal $\alpha_\star = R_0$. For all baselines (Standard CP, Cost-Aware CP, CRC), we calibrate their internal thresholds in the usual way across a pre-defined grid of possible $\alpha$ values.

2. **Select Operating Points on $\mathcal{D}_{\text{val}}$.** For each baseline, we evaluate its performance for every $\alpha$ in the grid on the validation set. We then compute the realized non-coverage risk, $\widehat{R}_{NC}^{(\text{val})}$, for each $\alpha$. We select the specific $\alpha^\dagger$ that results in a validation risk closest to our target $R_0$. CNCRC requires no tuning in this step as its threshold $q$ is already fixed.

3. **Report on $\mathcal{D}_{\text{test}}$.** With the parameters now frozen for all methods ($q$ for CNCRC and $\alpha^\dagger$ for each baseline), we perform a final evaluation on the held-out test set and report all metrics.

This protocol ensures that any observed differences in $\widehat{\text{AmbCost}}$ or other efficiency metrics are due to the methods' inherent properties, rather than a mismatch in their operating points.

## F.3 ANALYSIS OF SCORE DISTRIBUTIONS

Figure 2 provides a deeper, mechanistic explanation for the performance trade-offs observed in the main text's risk-risk plot (Figure 1). It visualizes the empirical distributions of the calibration scores for each method when aligned at a non-coverage risk of $\widehat{R}_{NC} \approx 0.08$. Each panel marks the calibrated threshold $q$ (red dashed line) and the sample mean of the scores (blue dotted line).

The distributions reveal how CNCRC achieves its superior efficiency. **CNCRC-Sum** (top-left) produces a broad, low-mean distribution with a relatively right-shifted threshold, yielding a large acceptance region; this allows it to include cost-safe candidates while suppressing risky ones, hence achieving the lowest ambiguity cost. **CNCRC-Max** (top-right) pushes most scores towards zero, creating a sharp separation between safe and unsafe candidates, which reflects its conservative nature. In contrast, **Standard CP** (bottom-left) concentrates scores at high values. Its cost-blind thresholding cannot distinguish dangerous from benign distractors, explaining its higher ambiguity cost. **Cost-aware CP** (bottom-right) improves slightly, but its distribution remains tightly centered and less risk-informative than CNCRC's. These distributions make the mechanism explicit: CNCRC reshapes the score space to reflect asymmetric risk, which is why its trade-off curve lies closest to the origin.

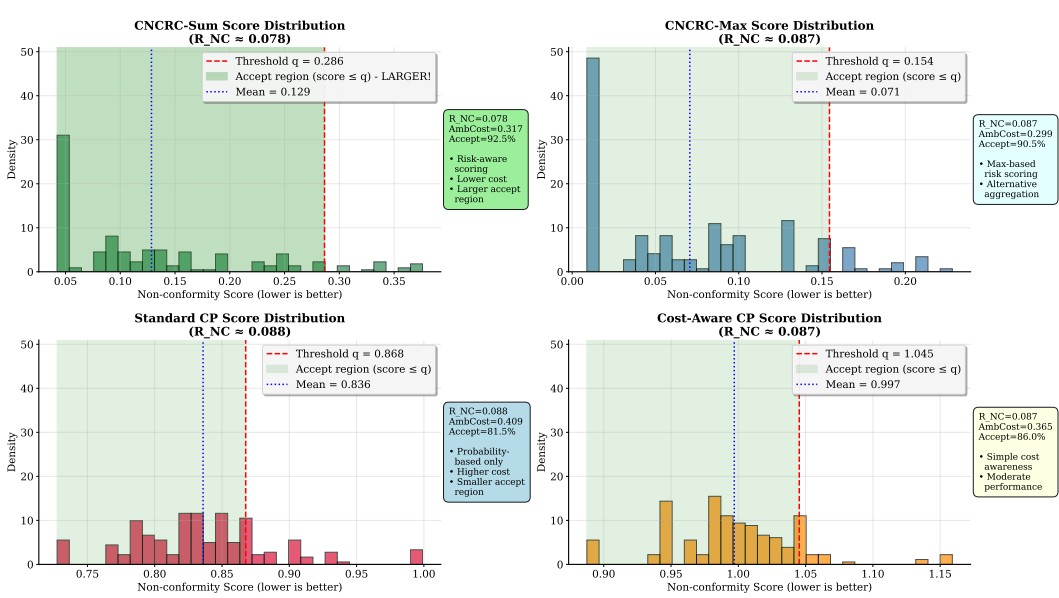

Figure 2: Score distributions at aligned $\widehat{R}_{NC} \approx 0.08$. CNCRC variants produce risk-aware score shapes that create a larger and safer acceptance region compared to the baselines, explaining their superior ambiguity costs.

