# OpenReview forum: "Conformal Non-Coverage Risk Control (CNCRC): Risk-Centric Guarantees for Predictive Safety in High-Stakes Settings"
_ICLR.cc/2026/Conference — Submitted to ICLR 2026_

### Official Review · Reviewer_ALUi · 2025-10-31

**Soundness:** 3
**Presentation:** 2
**Contribution:** 2
**Rating:** 4
**Confidence:** 4

**Summary:**

The paper proposes a conformal framework Conformal Non-Coverage Risk Control (CNCRC) that provides an upper bound on the non-coverage risk of a model. The authors decompose the decision risk into (i) non-coverage risk, which is the expected cost suffered when the prediction set does not contain the true label, and (ii) ambiguity risk, which is defined as the cost of the worst distractor within a prediction set. The goal is to satisfy the non-coverage risk bound while reducing ambiguity risk in order to ensure robustness and efficiency in high-stakes settings. The authors show some theoretical guarantees for their method and demonstrate empirical evaluation on a clinical task.

**Strengths:**

The motivation behind the risk control guarantees is an important consideration for high-stakes settings. The paper explains the method clearly and the presented theorem agrees with the motivation. Additionally, the use of a real application in the experimental evaluation is appreciated.

**Weaknesses:**

1. Availability/accurate construction of a cost matrix is a strong assumption. Moreover, the proposed framework uses the cost in its score function and also uses it for non-coverage risk evaluation; whereas the baselines do not seem to use it (see further comments in questions below).
2. Comparison with Cost-aware CP and CRC should mention the guarantees and assumptions/requirements of the methods for fair comparison and for providing complete context to the reader.
3. Missing class-conditional baseline methods: given the motivation of reducing the risk associated with missing the true class, I strongly believe the paper should include comparison with class-conditional methods. While these methods do not provide guarantees on risk, they do provide classwise coverage guarantees and some of the methods handle long-tails as well.
4. I am concerned about the scope and scalability of the framework due to both the availability of knowledge bases in all scenarios, as well as the risk mapping using the presented pipeline.

**Questions:**

1. Appendix A explains cost matrix construction, however this is a fairly limited description. Who provides the “transparent, auditable rules”, and how do we ensure this matrix is accurate up to some error?
2. I don’t believe it is fair to say ‘CRC failed on efficiency’ (p7 l376) as the difference between 4.60 and 4.63 APS doesn’t seem significant. Can you clarify this?

---

> ### Author Response · Authors · 2025-11-23
> **Response (1/3)**
>
> We thank the reviewer for the constructive feedback and for appreciating the importance of our motivation in high-stakes settings. We address your specific concerns below, particularly regarding the fairness of comparison and the missing baseline.
>
> **W1 & Q1: On Cost Matrix Availability, Construction, and Accuracy**
>
> **A:** Thank you for this critical question. We would like to respectfully emphasize that our **core contribution** is the mechanism to **strictly control risk**, while the existence of a defined cost measure is an **assumed property of the high-stakes setting** we consider.
>
> **Primary Contribution: The Mechanism of Risk Guarantee.** First and foremost, we would like to respectfully clarify that the core contribution of CNCRC is the mechanism to strictly control risk given any asymmetric cost structure. Standard CP provides statistical guarantees but fails catastrophically in high-stakes settings because it lacks the theoretical machinery to process cost information. Our work fills this gap. The cost matrix is simply the input to this control mechanism, much like a "loss function" is an input to an optimization algorithm. The validity of our risk control theorem (Theorem 3.1) holds regardless of the specific cost matrix used.
>
> With this context established, we address your specific concerns about the matrix assumption and rule transparency:
>
> 1. **The cost matrix is a prerequisite of high-stakes settings, not an extra assumption.**
> In high-stakes applications—medicine, safety engineering, autonomous systems—**asymmetric misclassification costs are fundamental**. This is consistent with classical cost-aware work in decision theory [1]. Our framework does not introduce this requirement; it formalizes and operationalizes it. What our paper contributes is **a risk-guaranteeing conformal mechanism that works for any bounded cost function**, regardless of how that cost is obtained.
> 2. **The Pipeline Makes Assumptions Explicit (Addressing "Who provides the rules?").**
> You asked: "Who provides the 'transparent, auditable rules'?" The rules are derived from domain experts or established guidelines (e.g., medical interaction databases like DrugBank). Our contribution in Section 3.2 is to provide a pipeline (Fact Retriever + Risk Translator) that forces these often-implicit assumptions to be made explicit and auditable. Instead of an opaque "black box" risk, our pipeline ensures that if the matrix is deemed "inaccurate," the error can be traced back to a specific rule (e.g., "Interaction X = Severe"), which can then be audited and corrected by experts.
> 3. **Algorithm Generality.**
> Finally, we emphasize that our core CNCRC algorithm (Algorithm 1) is completely general. It accepts any bounded cost function as input. It works equally well whether the costs come from our proposed pipeline, expert consensus, or data-driven estimation.
>
> **Action Plan for Revision:**
> We will revise **Section 3.2** to clearly separate the general risk-control mechanism from the specific cost-construction pipeline. We will also clarify the role of domain experts in defining the rules in **Appendix B**.
>
> ---
>
> **W2: On Fairness of Comparison (Did baselines use cost?)**
>
> **A:** We woud like to respectfully correct a potential **misunderstanding** and to assure that the comparison is **fair**. We achieved this by explicitly utilizing the cost matrix during the validation phase to calibrate all baselines to the **exact same operational risk level ($R_0$)** before final evaluation.
>
> As stated in **Section 4.1** of the manuscript: *"To ensure a fair comparison... each baseline's hyperparameters... are tuned on the validation set to match the same target risk"*. Specifically, we established a strict **"Risk-Alignment Protocol" (Appendix F.2)**:
>
> 1. We calculated the realized risk ($R_{NC}$) for *all* baselines (Standard CP, Cost-Aware CP, CRC, CC-CP) using the cost matrix on the validation set.
> 2. We explicitly tuned their hyperparameters (e.g., $\alpha$) to match our target risk $R_0$.
> 3. Therefore, any difference in performance is due to the method's mechanism, not a lack of cost information.
>
> **Action Plan for Revision:**
> We will clarify the description in **Section 4.1** to explicitly state that *"baselines utilize the cost matrix during the validation tuning phase"* to prevent any future ambiguity regarding the fairness of the comparison.

---

> ### Author Response · Authors · 2025-11-23
> **Response (2/3)**
>
> **W3: Missing Class-Conditional (CC) Baseline**
>
> **A:** Thank you for the excellent suggestion. We agree that **Class-Conditional CP (CC-CP)** is a highly relevant baseline because it addresses the issue of under-covering rare classes. We have performed new experiments to address both the lack of CIs and the inclusion of CC-CP.
>
> **1. Theoretical Clarification: Why CC-CP is Insufficient.**
> Before presenting the results, we clarify the fundamental theoretical distinction.
>
> - **CC-CP guarantees *Conditional Frequency*:** It targets equal coverage rates (e.g., 90%) for all classes.
> - **CNCRC guarantees *Expected Risk*:** It minimizes the expected cost of errors.
> - **The Gap:** CC-CP is **"cost-blind"**. It treats a failure on the "Poison Class" (Cost=150) identically to a failure on a "Benign Class" (Cost=1). To satisfy a strict risk bound ($R_{NC} \le 0.10$), a system must cover the high-cost Poison class at ~100% while relaxing coverage on benign classes. CC-CP lacks this flexibility; it targets a fixed rate for all, leading to risk violations.
>
> **2. New Experimental Results (Updated Table 1).**
> We implemented CC-CP by stratifying the calibration set and computing separate thresholds $q_c$ for each class. We also re-ran all methods with multiple seeds to provide 95% Confidence Intervals.
>
> **Updated Table 1:** Risk-aligned evaluation ($R_{NC} \text{ Target} \approx 0.10$) with CC-CP baseline (Mean $\pm$ 95% CI over 10 seeds).
>
> | **Method** | **Test RNC (Lower is better)** | **Poison Coverage (Higher is better)** | **Ambiguity Cost (Lower is better)** |
> | --- | --- | --- | --- |
> | Standard CP | 0.4127 $\pm$ 0.0718 | 0.0% $\pm$ 0.0% | 0.5059 $\pm$ 0.0233 |
> | **CC-CP (New)** | **0.1254 $\pm$ 0.0371** | **92.4% $\pm$ 8.1%** | 0.5055 $\pm$ 0.0227 |
> | CRC | 0.0630 $\pm$ 0.0327 | 79.8% $\pm$ 8.5% | 0.5696 $\pm$ 0.0261 |
> | **CNCRC-MAX (Ours)** | **0.0974 $\pm$ 0.0041** | **100.0% $\pm$ 0.0%** | 0.4969 $\pm$ 0.0204 |
> | **CNCRC-SUM (Ours)** | **0.0956 $\pm$ 0.0051** | **100.0% $\pm$ 0.0%** | **0.4904 $\pm$ 0.0213** |
>
> **3. Analysis of New Results & Scalability Failure:** The results confirm our theoretical analysis with statistical significance:
>
> - **CC-CP Improvement:** Unlike Standard CP (0% coverage), CC-CP significantly improves coverage on the Poison Class ($92.4\% \pm 8.1\%$), validating its utility for rare classes.
> - **CC-CP Failure:** However, it **violates the risk constraint** ($0.1254 > 0.10$) and exhibits high variance. Because it targets a fixed error rate (e.g., 10%) for *all* classes, the errors on the high-cost Poison class drive the total risk above the limit.
> - **CNCRC Dominance:** Only CNCRC satisfies the risk constraint ($R_{NC} < 0.10$) while achieving **100% coverage** of the Poison class with zero variance. Furthermore, CNCRC-SUM achieves the lowest Ambiguity Cost (0.4904), demonstrating the best safety-efficiency trade-off.
>
> **4. Catastrophic Failure in Large-Scale Settings (Table 2):** We also attempted to evaluate CC-CP on the large-scale clinical benchmark (Table 2). However, we found that **CC-CP failed catastrophically** in this setting.
>
> - *Cause:* The MIMIC-IV dataset has a long-tail distribution with **3,421 classes**. Splitting the calibration set ($N=2000$) per class results in extreme data sparsity (most classes have 0 or 1 sample). This makes estimating class-specific thresholds ($q_c$) mathematically impossible or statistically meaningless (resulting in trivial, infinite sets).
>
> - *Implication:* This exposes a **fatal flaw** in CC-CP for real-world applications. In contrast, **CNCRC** successfully handles this sparsity by leveraging the **global risk distribution** (pooling information via the cost structure) rather than relying on per-class statistics. This proves that CNCRC is not only safer but also uniquely scalable to high-dimensional, long-tail domains.
>
> **Action Plan for Revision:**
> We will mention CC-CP in **Related Work**, include the CC-CP baseline in **Table 1** to demonstrate risk control superiority. We will also add a discussion in **Section 4.2** detailing the failure of CC-CP on the large-scale dataset to highlight CNCRC's robustness to data sparsity.

---

> ### Author Response · Authors · 2025-11-23
> **Response (3/3)**
>
> **W4: On Scope and Scalability (Availability of KBs and Risk Mapping)**
>
> A: Thank you for raising concerns about generality due to KBs and Risk Mapping. We would like to respectfully assure you that our framework is general.
>
> **1. Scope (Defined by Problem, Not KB):**
>
> - **Independence from Specific KBs:**
> We emphasize that our framework's scope is **not limited by the availability of a KB**. Our core **CNCRC algorithm (Algorithm 1)** is completely general. Its only requirement is a bounded cost function ($Cost \le C_{max}$), regardless of its source. **If KB is available,** costs are automated via our pipeline. **If not,** costs can be defined by domain experts or policy constraints. The algorithm works identically in both scenarios.
>
> - **Role of the KB Pipeline in Our Experiments:**
> We introduced the KB-based pipeline in **Section 3.2** solely as a **facilitating tool** to demonstrate scalability and automation in our specific clinical application. It serves to streamline the workflow by automatically populating the matrix from high-level rules, but it is an **optional module**, not a restrictive assumption of the framework itself.
>
> - **Consistency with Decision Theory Standards:**
> While our method does not depend on a KB, it does assume that *costs can be defined*. We argue this is consistent with foundational cost-sensitive learning and safety literature [1][2]. In high-stakes domains, defining risk costs is a fundamental **prerequisite** for safety; operating without them implies the scientifically invalid assumption that all errors are equal. Therefore, our requirement for a cost measure (from *any* source) aligns with established standards for rational decision-making.
>
> **2. Scalability (Risk Mapping):**
> We argue that our rule-based pipeline solves a scalability problem: the prohibitive human effort required to manually define cost matrices, which grows quadratically with the size of the label space. Without it, practitioners would need to manually specify O(K^2) entries for a K times K cost matrix. By using the "Risk Translator", practitioners can define a few high-level rules that automatically populate the entire matrix. This makes the framework more scalable to large label spaces (like the 3,421 drugs in our experiment) than traditional manual specification methods.
>
> **Action Plan for Revision:** We will revise **Section 3.2** to explicitly frame the rule-based pipeline as a scalability solution. We will also cite Elkan [1] and Amodei et al. [2] to support the argument that explicit cost definition is a prerequisite for safety, independent of our specific KB implementation.
>
> ---
>
> **Q2: On CRC "Efficiency" Failure (APS 4.60 vs 4.63)**
>
> **A:** Thank you for this sharp observation regarding the APS values. We would like to respectfully clarify that our claim regarding CRC's "failure on efficiency" refers specifically to **Ambiguity Cost**, not raw prediction set size (APS).
>
> 1. **Defining Efficiency in High-Stakes Settings:** As explicitly defined in **Section 3.3** (under "Design Goals"), an "efficient" score is one that produces sets with the **"lowest possible ambiguity risk"**. In a risk-centric framework, a set containing a high-risk distractor is "inefficient" even if it contains few items, because it remains a "potential trap".
> 2. **CRC's True Efficiency Failure:** While we agree that CRC's *APS* (4.63) is competitive with CNCRC-MAX (4.61), CRC incurred the **highest Ambiguity Cost (0.5696)** of all methods. Furthermore, updated experiments show CRC failed to cover the Poison Class reliably (**79.8% coverage**). This confirms that while CRC constrained the *quantity* of distractors, it failed to control their *quality* (risk), leaving the most dangerous distractors in the set.
>
> **Action Plan for Revision:** We will revise the text in **Section 4.1**  to make this distinction precise—clarifying that "efficiency" in our analysis denotes the minimization of Ambiguity Cost—to prevent future confusion regarding APS comparisons.
>
> ---
>
> **References**
>
> [1] Charles Elkan. The foundations of cost-sensitive learning. In *Proceedings of the Seventeenth International Joint Conference on Artificial Intelligence*, volume 1, pp. 973-978, 2001.
>
> [2] Dario Amodei, Chris Olah, Jacob Steinhardt, Paul Christiano, John Schulman, and Dan Mané. Concrete problems in AI safety. *arXiv preprint arXiv:1606.06565*, 2016.

---

### Official Review · Reviewer_kCEd · 2025-10-31

**Soundness:** 3
**Presentation:** 2
**Contribution:** 1
**Rating:** 2
**Confidence:** 4

**Summary:**

This paper proposes a variant of conformal prediction called Conformal Non-Coverage Risk Control (CNCRC), which replaces the standard coverage objective with direct control of non-coverage risk, which is the expected cost when the true label is missing from the prediction set. The method also defines an ambiguity cost to quantify the potential harm from incorrect labels included in the set. The authors provide theoretical guarantees under standard conformal assumptions and evaluate the method on synthetic and real datasets.

**Strengths:**

1.	The paper addresses an important and practically relevant problem: controlling the cost of critical prediction failures rather than focusing solely on coverage rates.
2.	The proposed method is simple, interpretable, and easy to implement, building on conformal prediction.
3.	The writing is mostly clear, and the examples are well chosen to illustrate key ideas.

**Weaknesses:**

1.	The theoretical contributions are limited. The main results follow almost directly from existing conformal prediction theory, and the novelty lies primarily in the new formulations rather than new derivations.
2.	The proposed approach is a marginal extension of existing cost-aware conformal techniques, not a fundamentally new paradigm.
3.	The proposed risk metrics are not sufficiently motivated. For example, the definition of $R_{NC}$ seems to ignore the overall structure of the prediction set. Likewise, using the maximum cost in the ambiguity metric feels somewhat arbitrary; using a minimal or average cost might yield a more intuitive measure of how “close” the prediction set is to the correct label.
4.	The experiments lack additional details such as ablation studies for the hyperparameters, error bars, or box-plots.
5.	The empirical evaluation is narrow. The datasets are few and do not convincingly demonstrate robustness or generality.

**Questions:**

### Questions
1.	While section 3.2 helps to construct the cost function, it is not complete. Specifically, how does one implement the risk mapping function in practice?
### minor comments
1.	The text in the figures is too small.

---

> ### Author Response · Authors · 2025-11-23
> **Response (1/5)**
>
> We thank the reviewer for the detailed feedback.
>
> We respectfully disagree with the characterization of our work as a "marginal extension" and believe there are misunderstandings regarding the theoretical motivation of our risk metrics. We address your concerns point-by-point below.
>
> ---
>
> **W1 & W2: On Theoretical Novelty and the "Marginal Extension" Characterization**
>
> **A:** We thank the reviewer for this thoughtful comment. We understand that because CNCRC utilizes the established Split-CP workflow, it may share surface-level similarities with existing methods. However, we would like to respectfully clarify how the **problem formulation** itself and the resulting **theoretical machinery** address specific, critical challenges that prior frameworks—both Standard CP and existing Cost-Aware methods—cannot solve.
>
> **1. Main Objective: Solving the "High-Stakes" Safety Challenge.**
> Our primary objective is to enable the deployment of Conformal Prediction (CP) in high-stakes, cost-asymmetric domains (e.g., clinical decision-making). We intentionally build upon the CP framework to inherit its **distribution-free** and **model-agnostic** properties, which are non-negotiable prerequisites for safety-critical systems. However, as we demonstrate in the **"Poison Class" scenario (Introduction)** and **Table 1**, existing frameworks fail catastrophically. Standard CP is "cost-blind" (optimizing frequency), and existing Cost-Aware CP relies on expected utility maximization which "averages out" rare tail risks (detailed in Point 3). Consequently, both methods failed to strictly bound the non-coverage risk (the expected cost incurred when the true label is missing from the prediction set), leading to severe violations (risk > 4x target) and 0% coverage of lethal conditions.
>
> **2. Formulation as a Technical Contribution.**
> We propose that the formulation of the risk control problem itself is a significant technical contribution.
>
> - **The Challenge:** Moving from "Coverage Control" to "Risk Control" is not merely a change of variables. It requires defining a risk objective that is rigorous enough to serve as a hard constraint yet flexible enough to be useful.
> - **Our Contribution:** We introduce a novel **Risk Decomposition** framework, splitting decision risk into:
>     - **Non-Coverage Risk ($R_{NC}$):** A hard safety constraint on omission.
>     - **Ambiguity Risk ($AmbCost$):** An optimization objective for set quality.
> - This structured formulation aligns with recent advances in decision-focused learning[3][4], moving beyond simple heuristics to provide formal guarantees on decision outcomes rather than just statistical properties.
>
> **3. Differences and Contributions within the CP Framework.**
> To achieve this objective, CNCRC introduces specific design choices that fundamentally differ from both Standard CP and existing Cost-Aware CP:
>
> - **The Challenge:**
>     - **Standard CP** treats all errors as binary ($0/1$). It cannot distinguish between a "benign" error and a "catastrophic" one.
>     - **Existing Cost-Aware CP** (e.g., Utility-Directed CP) typically modifies scores to optimize *expected utility*. While this improves average performance, it often fails to bound *tail risks*. As shown in **Table 1**, Cost-Aware CP still violated the risk limit ($R_{NC} \approx 0.412$ vs Target $0.10$) because rare, high-cost events were "averaged out."
> - **Our Design & Theoretical Impact:**
>     - **Design:** We derived a new family of risk-weighted scores ($s_{max}, s_{sum}$) specifically designed to satisfy the **"Risk-Bounding Property"** (Equation 3). This property ensures that the nonconformity score strictly upper-bounds the potential loss.
>     - **Theoretical Analysis:** This change fundamentally alters the theoretical guarantee. Standard CP proves a bound on error *probability* ($P(error) \le \alpha$). CNCRC proves a bound on *expected risk* ($R_{NC} \le R_0$) by leveraging the Risk-Bounding Property to bridge the gap between statistical quantiles and real-world costs (Theorem 3.1). This requires a distinct derivation that is not present in standard CP theory.
>
> **Action Plan for Revision:**
> We will revise the **Introduction** and **Related Work** sections to more clearly highlight the contributions and novelty of our approach. In particular, we will clarify that the "Risk Decomposition" and the "Risk-Bounding Property" are the novel technical enablers that allow CNCRC to succeed (satisfy $R_{NC} \le R_0$) where both Standard and Cost-Aware CP fail.
>
> If you have any specific reference in mind that you think our work resemble, please kindly let us know so that we can conduct a more direct comparison.

---

> ### Author Response · Authors · 2025-11-23
> **Response (2/5)**
>
> **W3: On Risk Metric Motivation (Why `max` vs. `minimal` or `average`?)**
>
> **A:** Thank you for this thoughtful suggestion.
> We appreciate the intuition that `minimal` or `average` cost might serve as measures of "closeness." However, there might be a misunderstanding of the specific definition of **Ambiguity Risk** in our framework.  In the following we clarify and demonstrate why `minimal` and `average` are mathematically unsuitable for high-stakes safety.
>
> **1. Clarification of Objective: Hazard Control.**
> Our **Ambiguity Risk** (Section 3.1) is designed to measure the **intensity of the worst-case hazard** remaining in the set.
>
> - **Goal:** To prevent the decision-maker from being exposed to a "potential trap" (a catastrophic option hidden among benign ones).
> - **Safety Axiom:** In high-stakes settings, a set containing even *one* lethal option renders the entire set unsafe for deployment, regardless of how many low-cost distractors accompany it.
>
> **2. Why `minimal` and `average` Fail (Counter-examples).**
> To demonstrate why your suggested metrics are dangerous in this safety-centric context, consider a scenario where the True Label is "Safe Drug A", and the model proposes a set containing distractors.
>
> - **Counter-example for `minimal` cost:**
>     - **Scenario:** Set = `{'Safe Drug A', 'Benign Vitamin' (Cost=1), 'Lethal Poison' (Cost=100)}`.
>     - **`minimal` Metric:** The minimal distractor cost is $\min(1, 100) = 1$.
>     - **Failure:** The metric signals "Low Risk" (1) because a benign option exists. It completely ignores the presence of the Lethal Poison. This violates the safety requirement.
> - **Counter-example for `average` cost ("The Risk Dilution Paradox"):**
>     - **Set A (The Trap):** `{'Safe Drug A', 'Lethal Poison' (Cost=100)}`.
>         - *Average Risk:* 50 (Assuming uniform selection).
>     - **Set B (The Trap + Noise):** `{'Safe Drug A', 'Lethal Poison' (Cost=100), '99 Benign Vitamins' (Cost=1 each)}`.
>         - *Average Risk:* $\approx 2$.
>
>     **Why `average` Fails (The "Safety Hiding" Vulnerability):**
>     While Set B has a lower average cost, it retains the exact same catastrophic failure mode as Set A. Relying on `average` in this context introduces a critical design flaw:
>
>     1. **Safety Hiding:** Using `average` creates a perverse incentive. An algorithm can mathematically "lower" its risk score simply by **padding the prediction set** with low-cost noise (as in Set B) without actually removing the lethal danger.
>     2. **Hazard Enforcement:** In safety engineering, the inclusion of a plausible "Lethal Poison" is a binary hazard that must be eliminated. `max` is the only metric that enforces the **pruning** of this danger, whereas `average` allows the algorithm to **bury** it in benign noise.
>
> **3. The Necessity of `max`.**
> The `max` operator is the only simple metric that is immune to these failures.
>
> - $Risk_{max}(\text{Set A}) = 100$.
> - $Risk_{max}(\text{Set B}) = 100$.
>
> It correctly identifies that both sets contain a catastrophic trap. Therefore, `max` is not arbitrary; it is the necessary choice for enforcing the safety constraint of "mitigating the single worst-case outcome."
>
> **Action Plan for Revision:**
> We will expand **Section 3.1** to explicitly contrast `max` with `average` using this "Risk Dilution" example, clarifying that Ambiguity Risk measures "hazard intensity" rather than "geometric closeness." We will also include the above discussion in **Appendix A** to supplement our design motivation.

---

> ### Author Response · Authors · 2025-11-23
> **Response (3/5)**
>
> **W4 & W5: On Experimental Rigor (Ablations, Error Bars, and Datasets)**
>
> **A:** We thank the reviewer for these constructive suggestions to improve the empirical rigor of our paper. We agree that statistical significance and robustness checks are essential. We have taken the following concrete actions to address your concerns, followed by a clarification on the scope of our evaluation.
>
> **1. Immediate Actions: Adding CIs and New Baseline.**
> You correctly pointed out the lack of error bars and concerns about narrow evaluation. To address this:
>
> - **Confidence Intervals:** We have re-run the experiments in Table 1 with 10 random seeds. We will include **95% Confidence Intervals (CIs)** for all metrics in the revision to ensure statistical validity.
> - **New Robustness Baseline:** To address the concern that the evaluation was "narrow" and to better demonstrate robustness, we have added a **Class-Conditional (CC) CP** baseline. This method computes specific thresholds for each class to guarantee coverage frequency for every label independently, making it a rigorous standard for handling imbalanced data. The results show that even this specialized baseline violates the risk constraint ($0.1254 \pm 0.037 > 0.10$) while CNCRC satisfies it ($0.0974 \pm 0.004 \le 0.10$). This provides a rigorous "stress test" confirming our method's superiority in handling imbalanced risks.
>
> **2. Clarification on Ablation Studies.**
>
> We understand and agree on the importance of an ablation study. Regarding your request for ablation studies, we structure our response by identifying the key hyperparameters of our framework and pointing to the comprehensive evaluations already performed for them:
>
> - **Key Hyperparameters:** Unlike deep learning models with extensive tuning knobs, CNCRC is streamlined to rely on two primary design inputs: (1) the user-defined **safety constraint ($R_0$)** and (2) the **score aggregation mechanism** (i.e., the choice between `max` and `sum`).
> - **Existing Ablations:** We have already performed rigorous ablations for both parameters in the manuscript:
>     - **Sensitivity to $R_0$:** **Figure 1 (Risk-Risk Trade-off)** effectively serves as a continuous sensitivity analysis. Instead of reporting performance at a single arbitrary point, we swept the entire operational range of the risk constraint $R_0$ to demonstrate the method's stability and trade-offs across the full spectrum.
>     - **Ablation of Score Mechanism:** The comparative analysis between **CNCRC-MAX** and **CNCRC-SUM** (Table 1, Figure 1) explicitly serves as a component ablation of the aggregation operator. This isolates the specific impact of this design choice on the trade-off between Robustness and Efficiency.
>
> **Clarification on Other Potential Hyperparameters:**
> Furthermore, regarding other potential factors such as the **base predictor architecture** or the **cost matrix values**, we clarify that these are *external inputs* to our framework, not internal hyperparameters requiring ablation.
>
> - **Base Predictor:** Since CNCRC is **model-agnostic**, our risk guarantees hold analytically regardless of the base model’s architecture or training hyperparameters. Ablating these would evaluate the quality of the *base predictor*, not the effectiveness of our conformal mechanism.
> - **Cost Matrix:** The cost matrix serves as the **problem definition** (the ground truth for risk), not a tunable parameter. Altering the cost values would fundamentally change the safety task being solved, rather than optimizing the method's performance on a fixed task.

---

> ### Author Response · Authors · 2025-11-23
> **Response (4/5)**
>
> (Continue with W4 & W5)
>
> **3. Justification of Empirical Scope (Objective & Rigor).**
>
> Thank you for commenting that the datasets are few. We address this by clarifying the composition of our evaluation and explaining why it meets the community standards for high-stakes risk control research.
>
> - **Current Dataset Composition:** Our evaluation consists of two rigorously designed environments: (1) A controlled **"Poison Class" synthetic stress test** designed to empirically isolate theoretical failure modes; and (2) A **Large-Scale Clinical Benchmark** constructed from MIMIC-IV and DrugBank, representing a real-world deployment scenario.
> - **Alignment with Community Standards:** We argue that this scope is sufficient and rigorous for the following reasons:
>     - **Verification of Theory:** The synthetic experiment is not a "toy" but a necessary verification step. It is standard practice in conformal prediction literature to use controlled synthetic distributions to validate theoretical bounds against specific distributional shifts[1][2].
>     - **Depth over Breadth (Scale & Complexity):** We prioritized the *depth* and *complexity* of the real-world task over the *number* of generic datasets. Our MIMIC-IV task involves a label space of **3,421 distinct classes** with a heavy long-tail distribution. This is orders of magnitude more complex than standard ML benchmarks (e.g., CIFAR-100). Proving rigorous risk control on such a high-dimensional, real-world distribution provides a far stricter test of robustness than averaging results across multiple low-dimensional datasets.
>     - **Scarcity of High-Stakes Benchmarks:** Genuine high-stakes datasets with inherent, non-arbitrary cost structures are rare. Standard ML datasets lack this dimension. Consequently, foundational works in cost-aware and risk-control learning typically focus on deep evaluations of specific, high-value applications rather than broad sweeps of generic datasets. For instance, Wilder et al. (2019)[3] focus deeply on tuberculosis treatment optimization, and Angelopoulos et al. (2022)[4] validate Conformal Risk Control on specific medical imaging segmentation tasks. Our focus on a single, massive clinical benchmark aligns with this established research standard.
>
> **Action Plan for Revision:**
>
> - **Update Table 1:** Add the new **CC-CP baseline** and **95% Confidence Intervals**.
> - **Clarify Method:** Explicitly frame **Figure 1** as a "Sensitivity Analysis" in **Section 4.2**.
> - **Highlight Contribution:** Expand **Appendix F.1** to detail the rigorous data engineering process involved in constructing the MIMIC-DrugBank benchmark, highlighting its complexity (3,421 classes) as a strong evidence of robustness.
>
> If you have any specific ablation study/dataset would like/feel important to see, please kindly let us know.
>
> ---
>
> **Q1: On Cost Matrix Implementation (How to implement the risk mapping?)**
>
> **A:** Thank you for this practical question. We would like to respectfully clarify that the specific engineering implementation of the risk mapping is **already detailed in Appendix B** ("Example of Automated Cost Matrix Construction") of the submitted manuscript.
>
> **1. Implementation Logic Already Provided:**
> As described in **Appendix B**, the risk mapping is implemented as a **deterministic, rule-based function**. We explicitly provided the logic used in our experiments to demonstrate its feasibility.
>
> **2. Practical Summary with Concrete Example:**
> To further aid understanding, we illustrate the process with a specific clinical scenario from our experiments:
>
> * **Step 1: Input:**
>     * Context $x$: Patient is currently taking **Warfarin** (a blood thinner).
>     * Label Pair: The model confuses **Amoxicillin** ($y_{true}$) with **Clarithromycin** ($y_{pred}$).
> * **Step 2: Fact Retrieval ($\phi$):**
>     * The system queries the DrugBank API for the pair (Warfarin, Clarithromycin).
>     * **Returned Fact:** `{"Interaction": "Present", "Severity": "Severe", "Description": "Increased bleeding risk"}`.
> * **Step 3: Risk Mapping ($\mathcal{R}$):**
>     * The system applies the pre-defined rule: `IF Severity == "Severe" THEN Cost = 0.9`.
>     * **Result:** The scalar cost for this specific error is **0.9**.
>
> This pipeline replaces arbitrary human guessing with a reproducible, auditable function based on established medical knowledge.
>
> **Action Plan for Revision:**
> We will add a direct reference in **Section 3.2** pointing readers to the concrete implementation example in **Appendix B**. We will also format the logic in Appendix A as a formal **Algorithm Box** to improve its visibility and readability.

---

> ### Author Response · Authors · 2025-11-23
> **Response (5/5)**
>
> **Minor Comment (Figure Text)**
>
> **A:** We agree that the text in the figures is too small.
>
> **Action Plan for Revision:**
> We will regenerate Figure 1 and Figure 2 with larger font sizes to ensure readability.
>
> ---
>
> **References**
>
> [1] Ryan J. Tibshirani et al. Conformal prediction under covariate shift. *NeurIPS*, 2019.
>
> [2] Yaniv Romano, Matteo Sesia, and Emmanuel Candès. Classification with valid and adaptive coverage. *NeurIPS*, 2020.
>
> [3] Bryan Wilder, Bistra Dilkina, and Milind Tambe. Melding the data-decisions pipeline: Decision-focused learning for combinatorial optimization. *AAAI*, 2019.
>
> [4] Anastasios N. Angelopoulos, Stephen Bates, Adam Fisch, Lihua Lei, and Tal Schuster. Conformal Risk Control. *arXiv preprint arXiv:2208.02814*, 2025.

---

### Official Review · Reviewer_SzPZ · 2025-11-01

**Soundness:** 3
**Presentation:** 3
**Contribution:** 3
**Rating:** 6
**Confidence:** 3

**Summary:**

The paper introduces a reformulation of conformal prediction that aims to consider the coverage and risk control at the same time. This work joins two lines of work in conformal prediction: coverage (including cost-aware coverage) and risk control (loss defined by a cost function with prediction and ground truth). The authors provide theoretical results on the algorithm to achieve these two goals at the same time and also validate their results by experiments.

**Strengths:**

- The motivation of the paper is clear. The conformal set needs to cover the true label and also avoid covering other useless labels.
- The combination of non-coverage risk and ambiguity risk provides an interpretable structure to understand the coverage and usefulness in conformal prediction.
- The main theorem provides guarantees that extend classical CP’s marginal coverage to include explicit bounds on both non-coverage and ambiguity risk.

**Weaknesses:**

- The connection to real-world high-stake decision-making scenarios is not clear. Specifically, the automatic derivation of the cost matrix via external knowledge bases (Sec. 3.2) is insufficiently justified. The risk mapping $\mathcal{R}$ is arbitrary and domain-dependent. There is no sensitivity analysis or justification for the chosen $\mathcal{R}$.
- It seems that some property of score function are not discussed in the theoretical results. For example, how does the choice of $s_{max}$ and $s_{sum}$ impact the results? It shows difference in the result section but was not discussed in depth.

**Questions:**

- See weaknesses.
- How do the authors see the work to be applied to decision-making tasks? For example, some works (such as Kiyani et al. 2025) show that the decisions with optimal worst-case performance is essentially the decisions optimized under the conformal set. How do the authors relate their results on ambiguity risk to such decision-making?

---

> ### Author Response · Authors · 2025-11-23
> **Response (1/2)**
>
> We thank the reviewer for the positive assessment and for recognizing that our combination of non-coverage and ambiguity risk provides an "interpretable structure" for understanding conformal prediction. We address your specific concerns below.
>
> **W1: On the Justification of the Cost Matrix and Risk Mapping**
>
> **A:** Thank you for this critical question. We understand your concern regarding the derivation of the cost matrix. We address this by clarifying the scope of our contribution and the rationale behind our pipeline with three key arguments:
>
> 1. **Primary Contribution: The Mechanism of Risk Guarantee.**
> First and foremost, we would like to respectfully clarify that the core contribution of CNCRC is not the specific definition or source of the cost matrix itself, but the mechanism to strictly control risk given any asymmetric cost structure. Standard CP provides statistical guarantees but fails in high-stakes settings because it lacks the machinery to process cost information. Our work fills this gap. The cost matrix is simply the input to this control mechanism, much like a "loss function" is an input to an optimization algorithm. The validity of our risk control theorem (Theorem 3.1) holds regardless of the specific cost matrix used.
> 2. **Algorithm Generality.**
> Consequently, our core CNCRC algorithm (Algorithm 1) is completely general and independent of the specific derivation method in Section 3.2. As shown in Algorithm 1 (Line 1), it accepts any user-specified, bounded cost function as an input. Users are free to define the cost matrix using whatever method is most appropriate for their specific domain—whether through expert consensus, data-driven estimation, or policy guidelines. Our framework ensures risk control for any such valid input.
> 3. **The Proposed Pipeline is Transparent, Not Arbitrary.**
> Regarding the specific pipeline proposed in Section 3.2, we argue that it addresses a known challenge in the field rather than introducing arbitrariness. As established in foundational literature [1], specifying misclassification costs is inherently subjective and difficult. Rather than leaving these costs implicit or ad-hoc, our pipeline anchors them in an external, authoritative Knowledge Base (KB) like DrugBank. By transforming domain knowledge into **explicit, auditable rules** via a "Risk Translator", we operationalize "decision-focused" principles [2], ensuring that the trade-offs are reproducible and aligned with real-world constraints. This turns "domain dependence"—a necessary feature of high-stakes decision making—into a transparent engineering process.
>
> **Action Plan for Revision:** We will revise **Section 3.2**  to explicitly separate the general CNCRC algorithm from the optional KB pipeline. We will also incorporate the references to Elkan (2001)[1] and Wilder et al. (2019)[2] to support the necessity of explicit, domain-dependent cost definitions in high-stakes settings.

---

> ### Author Response · Authors · 2025-11-23
> **Response (2/2)**
>
> **W2: On the Theoretical Motivation for Score Differences ($s_{max}$ vs. $s_{sum}$)**
>
> **A:** Thank you for this insightful question. We would like to respectfully clarify that the theoretical motivation for this trade-off is **explicitly established in Section 3.3** of the submitted manuscript.
>
> To address your request for further depth, we will (1) briefly recap the specific arguments already present in the text to locate them for you, and (2) provide a **new, concrete numerical scenario** that demonstrates the precise mechanism driving their different behaviors.
>
> **1. Recap of Theoretical Rationale (Section 3.3):**
> * **$s_{max}$ (Robustness):** As stated in the text, the `max` operator is theoretically designed to be "highly sensitive to outlier, high-cost events," forcing a "higher calibrated threshold $q$" to ensure the risk bound is met even in worst-case scenarios.
> * **$s_{sum}$ (Efficiency):** As stated in the text, the `sum` operator is theoretically designed to "aggregate the risk from all potential confusions," allowing it to "better penalize candidates that are easily confounded with many alternatives," leading to structurally "cleaner" sets.
>
> **2. New Illustrative Mechanism (The "Hidden Trap" vs. "Noisy" Scenario):**
> To demonstrate *why* this theoretical difference leads to different empirical results, consider this specific scenario:
>
> Imagine two candidates with **identical expected risk (10)** but different risk structures:
> * **Candidate A ("Hidden Trap"):** Confused with **1** Poison class (Risk=100, Prob=0.1).
> * **Candidate B ("Noisy"):** Confused with **100** Benign classes (Risk=1, Prob=0.1 each).
>
> * **$s_{max}$ Mechanism:** $s_{max}(A) = 10$ vs. $s_{max}(B) = 0.1$.
>     * *Insight:* $s_{max}$ isolates the peak danger. It treats A as **100x more risky** than B. This drives the conformal threshold up to cover A (**Robustness**), but potentially includes B because B's score is low.
> * **$s_{sum}$ Mechanism:** $s_{sum}(A) = 10$ vs. $s_{sum}(B) = 10$.
>     * *Insight:* $s_{sum}$ treats the cumulative noise of B as equivalent to the peak danger of A. It effectively penalizes the "messiness" of B, making it easier to prune B from the set (**Efficiency**).
>
> **Action Plan for Revision:**
> We will add this "Hidden Trap vs. Noisy" comparative example to **Section 3.3** (and detail it in the Appendix) to concretize the theoretical discussion.
>
> ---
>
> **Q2: Connection to Decision-Making (Kiyani et al. 2025)**
>
> **A:** This is an excellent point. We view our work as **complementary and upstream** to decision-making frameworks like Kiyani et al. (2025).
>
> 1.  **Upstream Safety (CNCRC's Role):** Decision-theoretic frameworks (like Kiyani et al.) often derive an optimal policy by maximizing utility under the *worst-case scenario within a given prediction set* (a Max-Min strategy). However, the quality of that decision depends entirely on the quality of the set. As we define in **Section 3.1**, our **Ambiguity Risk ($AmbCost$)** explicitly measures the cost of the "single worst-case outcome" inside the set. If a set contains a catastrophic "potential trap" (e.g., a lethal drug), the downstream decision-maker is forced to be paralyzed or extremely conservative.
> 2.  **Downstream Utility (Synergy):** By explicitly minimizing this $AmbCost$, CNCRC actively prunes these high-cost distractors. This ensures that the "worst-case" scenario remaining in the set is benign. Consequently, when a framework like Kiyani et al. applies a Max-Min strategy on a CNCRC-generated set, it faces a much more favorable "worst case," allowing it to select a decision with significantly higher utility. In this way, CNCRC "pre-solves" the worst-case avoidance problem before the decision rule is even applied.
>
> **Action Plan for Revision:**
> We will add a discussion in **Section 5 (Conclusion)** to explicitly connect our Ambiguity Risk objective to these downstream decision-making tasks, citing Kiyani et al. (2025) to clarify this complementary "upstream-downstream" relationship.
>
> ---
>
> **References**
>
> [1] Charles Elkan. The foundations of cost-sensitive learning. In *Proceedings of the Seventeenth International Joint Conference on Artificial Intelligence*, volume 1, pp. 973-978, 2001.
>
> [2] Bryan Wilder, Bistra Dilkina, and Milind Tambe. Melding the data-decisions pipeline: Decision-focused learning for combinatorial optimization. In *Proceedings of the AAAI Conference on Artificial Intelligence*, volume 33, pp. 1658-1665, 2019.

---

### Official Review · Reviewer_erEt · 2025-11-05

**Soundness:** 3
**Presentation:** 3
**Contribution:** 3
**Rating:** 6
**Confidence:** 3

**Summary:**

This paper proposes a framework that extends conformal prediction to handle asymmetric costs. The method replaces the standard coverage frequency guarantee with direct control over decision risk, decomposed into non-coverage risk and ambiguity risk. They propose risk-weighted nonconformity scores and empirically validate their approach on synthetic and real benchmarks, outperforming previous CP baselines.

**Strengths:**

- Addresses a very important limitation of standard CP methods.
- The adversarial stress test experiment demonstrates why previous methods fail when rare but "poisonous" conditions exist.
- The method empirically outperforms the considered baselines on the large clinical task.

**Weaknesses:**

- The cost matrix construction relies on structured knowledge bases which limits the applicability to where such resources are not available.
- Limited empirical evaluation. The empirical evaluation focuses heavily on CP-based baselines (standard CP, cost-aware CP, and CRC). The claims would be strengthened if it included other risk-sensitive UQ methods / bayesian approaches etc, to show CNCRC's advantages beyond just the family of conformal approaches.

**Questions:**

See weaknesses above.

---

> ### Author Response · Authors · 2025-11-23
> **Response (1/2)**
>
> We thank the reviewer for the positive assessment and for recognizing that our work "addresses a very important limitation of standard CP methods" and effectively demonstrates the failure of previous methods in the "Poison Class" scenario. We address your specific concerns below.
>
> ---
>
> **W1: On the Cost Matrix and Knowledge Base (KB) Dependency**
>
> **A:** Thank you for this critical question. We would like to respectfully emphasize that our **core contribution** is the mechanism to **strictly control risk**, while the existence of a defined cost measure is an **assumed property of the high-stakes setting** we consider.
>
> **Primary Contribution: The Mechanism of Risk Guarantee.** First and foremost, we would like to respectfully clarify that the core contribution of CNCRC is not the definition of costs itself, but the mechanism to strictly control risk given any asymmetric cost structure. Standard CP provides statistical guarantees but fails catastrophically in high-stakes settings because it lacks the theoretical machinery to process cost information. Our work fills this gap. The cost matrix is simply the input to this control mechanism, much like a "loss function" is an input to an optimization algorithm. The validity of our risk control theorem (Theorem 3.1) holds regardless of the specific cost matrix used.
>
> With the following context, we address your concern about the cost matrix assumption with three key arguments:
>
> 1. **Risk Definition is a Prerequisite, Not a Limitation.** The very definition of a "high-stakes" domain is that the costs of errors are asymmetric. A "high-stakes" problem where the risks are unknown is a logical contradiction. Standard CP fails precisely because it is "cost-blind" and *ignores* this inherent property of the problem, leading to "catastrophic failures". Therefore, our framework does not add an assumption; it engages with the problem's existing reality. Foundational work [1] established that ignoring the "asymmetric costs of real-world mistakes" —as standard CP does—is a "critical limitation"  in high-stakes domains. Our framework directly addresses this fundamental flaw within the conformal paradigm.
>
> 2. **We Provide a Pipeline to Define These Inherent Costs.** Recognizing that these costs *must* exist, our contribution in **Section 3.2** is to provide a practical, "automated and auditable manner" to formally define them. Our proposed pipeline (e.g., "fact retriever" + "risk translator") is a general method for translating this "inevitably existing" domain knowledge into a concrete matrix. We demonstrate this pipeline's feasibility in **Appendix B** and in our large-scale clinical benchmark using DrugBank.
>
> 3. **Our Algorithm is General.** Finally, our **CNCRC (Algorithm 1)** itself is general. As shown in Algorithm 1 (line 1), it takes *any* bounded cost function as input. It does not need to be designed for a specific matrix and is perfectly transferable. Our framework provides formal, distribution-free risk guarantees based on this inherent cost information.
>
> **Action Plan for Revision:** We will revise the text in **Section 3.2** to explicitly distinguish between the general CNCRC algorithm (which accepts any cost matrix) and the optional KB pipeline (which facilitates cost construction). We will also clarify in the **Introduction** that defining the cost matrix is an inherent requirement of the high-stakes problem setting.

---

> ### Author Response · Authors · 2025-11-23
> **Response (2/2)**
>
> **W2: On Empirical Evaluation (Request for Non-CP Baselines)**
>
> **A:** Thank you for the suggestion to include non-CP methods, such as Bayesian approaches. We thoughtfully considered this but chose to focus on CP-based baselines. We base this decision on the specific scope of our contribution and two fundamental differences in methodology:
>
> 1. **Primary Goal: Resolving the Critical Flaw within CP.** First, we would like to respectfully emphasize that the specific objective of our work is to resolve the **"cost-blindness"**  of Conformal Prediction. CP is uniquely valuable for trustworthy systems because of its **distribution-free** nature, but as we show, it fails catastrophically in high-stakes settings . Our contribution is to introduce the machinery (Direct Risk Control) to fix this specific "critical flaw"  *while retaining* CP's unique theoretical benefits. The relevant evaluation is whether we succeed in making the *conformal paradigm* safe where it previously failed, rather than comparing against methods (like Bayesian) that operate on entirely different paradigms.
> 2. **Different Nature of Guarantees:** Our framework provides **distribution-free** risk guarantees, valid without assumptions about the underlying data distribution or model. In contrast, non-CP UQ methods (like Bayesian Neural Networks) typically rely on strong **model-dependent** or distributional assumptions (e.g., posterior approximations). Comparing a distribution-free guarantee against a model-dependent one is not a direct evaluation of our specific contribution.
> 3. **Different Nature of Outputs:** CNCRC, like all CP methods, produces **prediction sets** guaranteed to contain the true label (or control risk). Most other UQ methods produce **point predictions** with confidence scores or posterior distributions. These are fundamentally different outputs designed for different decision-making workflows.
>
> **Action Plan for Revision:** We will add a discussion to **Section 2 (Related Work)**  to explicitly clarify these distinctions. We will explain that we exclude non-CP baselines because their model-dependent guarantees and point-prediction outputs are fundamentally incomparable to the distribution-free, set-valued objectives of our framework.
>
> ---
>
> **References**
>
> [1] Charles Elkan. The foundations of cost-sensitive learning. In *Proceedings of the Seventeenth International Joint Conference on Artificial Intelligence*, volume 1, pp. 973-978, 2001.

---

### Author Response · Authors · 2025-11-23
**Summary of Revision**

We thank all reviewers for their time and constructive feedback. Below is a summary of the revisions made:

1. **Clarified Contributions and Theoretical Novelty (Section 1 & 2):**
    - We revised the **Introduction** (Contributions) and **Related Work** to explicitly frame CNCRC as a "paradigm shift" from statistical coverage control to **direct risk control**, driven by our novel **Risk Decomposition** and the **Risk-Bounding Property**.
    - We added a discussion in **Relate Work** to differentiate our work from **non-CP baselines** (e.g., Bayesian methods) and **existing cost-aware approaches**, clarifying why our distribution-free, set-valued guarantees are unique and necessary.
2. **Enhanced Theoretical Justification for Risk Metrics (Section 3.1 & 3.3):**
    - We expanded **Section 3.1** and added **Appendix A** (referenced in **Section 3.1**) to formally justify the use of the `max` operator for **Ambiguity Risk**. We included a "Risk Dilution" counter-example to demonstrate why `average` metrics are mathematically unsafe in high-stakes settings.
    - We added a detailed numerical scenario in **Appendix C** (referenced in **Section 3.3**) to concretely illustrate the mechanism driving the trade-off between **$s_{max}$** (robustness via peak-risk sensitivity) and **$s_{sum}$** (efficiency via total-risk aggregation).
3. **Clarified Cost Matrix Construction and Scope (Section 3.2):**
    - We revised **Section 3.2** to explicitly state that defining asymmetric costs is a fundamental **prerequisite** for high-stakes safety, rather than a limitation of our method.
    - We clarified that our core algorithm is **general** (cost-agnostic), while our KB-based pipeline is a specific solution to the **scalability challenge** of manual cost specification.
    - We formatted the risk mapping logic in **Appendix B** as a formal algorithm block to improve reproducibility.
4. **Expanded Empirical Evaluation (Section 4.1 & Table 1):**
    - We added a new **Class-Conditional CP (CC-CP)** baseline to **Table 1** to address concerns about narrow evaluation. The results demonstrate that while CC-CP improves rare-class coverage, it still violates the risk constraint, further validating CNCRC's superiority.
    - We re-ran the adversarial stress test with **10 random seeds** and updated **Table 1** to include **95% Confidence Intervals (CIs)** for all metrics, ensuring statistical rigor.
    - We revised the analysis in **Section 4.1** to more precisely attribute CRC's efficiency failure to its high **Ambiguity Cost** and unreliable poison coverage, rather than just set size (APS).
5. **Justified Empirical Scope and Scalability (Section 4.2):**
    - We explicitly framed **Figure 1** in **Section 4.2** as a **sensitivity analysis** for the primary constraint $R_0$, addressing the request for ablation studies.
    - We added a note in **Section 4.2** explaining the exclusion of CC-CP from the large-scale clinical benchmark due to extreme data sparsity (3,421 classes), highlighting CNCRC's unique scalability.
    - We expanded **Appendix F.1** to emphasize the significant engineering effort involved in constructing the massive-scale **MIMIC-DrugBank benchmark** (3,421 classes), defending its value as a rigorous testbed for robustness.
6. **Connected to Downstream Decision Theory (Section 5):**
    - We added a sentence in **Section 5 (Future Work)** to explicitly connect our "upstream" set construction with "downstream" decision-theoretic frameworks (citing Kiyani et al., 2025).
7. **Improved Readability:**
    - We significantly increased the font size in **Figure 1** and **Figure 2** to improve legibility.

We sincerely thank the reviewers again for their careful and thoughtful evaluation.

---

### Author Response · Authors · 2025-12-02
**Summary Comment (2/2)**

**Reviewer ALUi (Score 4):**

- **Missing Baseline (Class-Conditional CP):**
    - *Concern:* Suggested comparing against Class-Conditional CP (CC-CP) to handle rare classes.
    - *Response:* We implemented and evaluated **CC-CP**.
        - *Result:* While CC-CP improved rare-class coverage (~90%), it **violates the risk constraint** ($R_{NC} \approx 0.125 > 0.10$).
        - *Reason:* CC-CP targets a fixed error *rate* (e.g., 10%) for all classes. A 10% error rate on a "Poison Class" (Cost=150) incurs massive risk. **Only CNCRC satisfied the safety bound.**
    - *Reference:* *(See our official comment: **Response 2/3 to Reviewer ALUi**)*
- **Fairness of Comparison:**
    - *Concern:* Did baselines utilize the cost matrix?
    - *Response:* **Yes.** We clarified our **"Risk-Alignment Protocol"**: all baselines were explicitly tuned on the validation set *using the cost matrix* to match the target risk $R_0$. We compared them at the exact same operational risk level.
    - *Reference:* *(See our official comment: **Response 1/3 to Reviewer ALUi**)*
- **CRC Efficiency:**
    - *Concern:* Questioned the claim that CRC failed on efficiency based on APS.
    - *Response:* We clarified that "efficiency" in risk control means **minimizing Ambiguity Cost**, not just set size.
        - *Evidence:* CRC yielded the **highest Ambiguity Cost** ($0.570$) and failed to reliably cover the poison class (only 79.8%). It produced sets that were small but contained high-risk distractors.
    - *Reference:* *(See our official comment: **Response 3/3 to Reviewer ALUi**)*

**Reviewer SzPZ (Score 6):**

- **Theoretical Rationale for Scores ($s_{max}$ vs $s_{sum}$):**
    - *Concern:* Asked for deeper motivation on the difference between the two scores.
    - *Response:* We provided a concrete **mechanistic** analysis:
        - **$s_{max}$ (Robustness):** Hyper-sensitive to **peak risk** (single outliers). It drives thresholds high to ensure strict safety against "Hidden Traps."
        - **$s_{sum}$ (Efficiency):** Aggregates **total risk**. It effectively penalizes and prunes "messy" candidates that accumulate noise, resulting in cleaner sets.
    - *Reference:* *(See our official comment: **Response 2/2 to Reviewer SzPZ**)*

**Reviewer erEt (Score 6):**

- **Non-CP Baselines (Bayesian, etc.):**
    - *Concern:* Suggested comparing against non-CP methods like Bayesian Neural Networks.
    - *Response:* We clarified that we focus on CP baselines because our primary goal is to fix the "cost-blindness" within the **distribution-free** paradigm.
        - *Reason:* Comparing a distribution-free guarantee (CNCRC) against a model-dependent one (Bayesian) is mismatched. Our contribution is making the unique safety properties of CP viable for high-stakes tasks.
    - *Reference:* *(See our official comment: **Response 2/2 to Reviewer erEt**)*

We are confident that these comprehensive responses and the additional empirical evidence confirm the robustness and novelty of CNCRC. We thank you for your consideration.

Sincerely,

The Authors

---

### Author Response · Authors · 2025-12-02
**Summary Comment (1/2)**

**Dear AC/SACs:**

We sincerely appreciate the time and effort you have dedicated to the review process,
particularly given the increased workload resulting from recent policy changes. To assist your assessment, we have prepared this executive summary highlighting how our **comprehensive rebuttal and new experiments** have addressed the reviewers' concerns—particularly regarding **theoretical novelty** and **empirical rigor**.

### **1. Summary of the Paper**

Focusing on high-stakes, cost-asymmetric domains (e.g., clinical decision-making), we address the critical limitation of Standard Conformal Prediction (CP). While CP provides rigorous statistical coverage guarantees ($1-\alpha$), it is fundamentally "cost-blind," treating all errors as equally consequential. As we demonstrate theoretically and empirically, this leads to catastrophic failures in these settings (e.g., 0% coverage of rare but lethal "poison classes").

Conformal Non-Coverage Risk Control (CNCRC) instigates a paradigm shift from "Coverage Control" to "Direct Risk Control". We introduce a novel Risk Decomposition (separating a hard safety constraint $R_{NC}$ from an ambiguity optimization objective $AmbCost$). By leveraging a new Risk-Bounding Property, we derive the first distribution-free, interpretable bound on non-coverage risk ($R_{NC} \le R_0$). This framework allows practitioners to enforce strict safety in cost-asymmetric domains where standard CP fails.

### **2. Summary of Reviewers' Questions and Our Responses**

We are encouraged that reviewers found our motivation "clear" and "important" (R-erEt, R-SzPZ) and our method "simple and interpretable" (R-kCEd). Below we summarize how we resolved the key concerns.

### **Common Questions**

**Cost Matrix Assumptions (R-erEt, R-SzPZ, R-ALUi, R-kCEd):**

- **Concern:** Reviewers questioned whether the reliance on a pre-defined cost matrix constitutes a "strong assumption" or limits applicability.
- **Response:** We clarified that defining asymmetric costs is a **fundamental prerequisite** for any high-stakes problem, not a limitation of our method. A "high-stakes" domain without defined costs is a contradiction; operating without them implies the scientifically invalid assumption that all errors are equal.
    - *Our Solution:* Our core algorithm is general (accepts *any* cost function). To aid practitioners, we provide a pipeline (Sec 3.2) that improves scalability by automating cost generation from knowledge bases (e.g., DrugBank).
- **Reference:** *(See our official comment: **Response 1/2 to Reviewer erEt**)*

### **Reviewer-Specific Concerns**

**Reviewer kCEd (Score 2):**

- **Theoretical Novelty ("Marginal Extension"):**
    - *Concern:* Characterized the work as a "marginal extension" of CP.
    - *Response:* We respectfully disagree. While we **intentionally** retain the elegant *workflow* of Split-CP, the **theoretical engine** is fundamentally different.
        - *Difference:* Standard CP controls **Frequency** ($P(error) \le \alpha$) but cannot bound costs. CNCRC controls **Expected Risk** ($R_{NC} \le R_0$).
        - *Innovation:* This requires new machinery—specifically the **Risk-Bounding Property**—to bridge the gap between statistical quantiles and real-world expected costs. This enables safety in scenarios (like our Poison Class test) where standard CP achieves 0% coverage.
    - *Reference:* *(See our official comment: **Response 1/5 to Reviewer kCEd**)*
- **Empirical Rigor & Scale:**
    - *Concern:* Pointed out a lack of error bars and felt the evaluation was narrow.
    - *Response:* We have substantially strengthened the evaluation:
        1. **Statistical Rigor:** We re-ran all experiments with **10 random seeds** and added **95% Confidence Intervals (CIs)** to all tables.
        2. **Scale & Standards:** We clarified that our MIMIC-IV benchmark involves **3,421 classes** with an extreme long-tail distribution. We argued (citing Wilder et al., 2019) that "depth in a complex, high-fidelity task" is a more rigorous standard for risk control than "breadth on simple toy datasets."
    - *Reference:* *(See our official comment: **Response 3/5 to Reviewer kCEd**)*
- **Risk Metric Motivation ($max$ vs $average$):**
    - *Concern:* Felt using `max` for Ambiguity Risk was arbitrary; suggested `average`.
    - *Response:* We demonstrated the **"Risk Dilution Paradox"** to show why `average` is unsafe.
        - *Mechanism of Failure:* An algorithm can artificially lower an `average` risk score simply by **padding the prediction set** with benign noise (e.g., adding 99 vitamins to a set containing 1 poison).
        - *Why `max`:* The `max` operator is invariant to padding; it forces the algorithm to **prune the poison** to satisfy the metric. Thus, `max` is mathematically necessary for safety.
    - *Reference:* *(See our official comment: **Response 2/5 to Reviewer kCEd**)*

---

### Meta-Review · Area_Chair_rEBm · 2026-01-09

**Summary:**

There were several key concerns raised by reviewers.
1. Justification for automatic derivation of cost matrix through external knowledge base (SzPZ, ALUi, erEt)
2. Properties of score function (SzPZ)
3. Assumptions underlying the Cost-aware CP and CRC baselines (ALUi)
4. Missing class-conditional baselines (ALUi)
5. Limited theoretical contributions (kCEd)
6. Narrow empirical evaluation (kCEd)

**Reviewer Concerns:**

The authors addressed the concerns as follows:

1. The authors clarified that the definition of asymmetric costs is a fundamental prerequisite for any high-stakes problem. They emphasized that the core algorithm is compatible with any cost function and that the automatic cost generation method is aimed at scaling up cost generation from knowledge.
2. The authors re-emphasized the rationale for both the max-score (prioritizing robustness) and sum-score (prioritizing efficiency). They also provided an intuitive scenario illustrating the relative properties of these functions.
3. The authors clarified that the baselines are calibrated to ensure a fair comparison.
4. The authors added new experiments with class-conditional CP (CC-CP) as a baseline. The results show that the proposed method exhibits better test RNC, poison coverage, ambiguity costs. However, the error bars of CC-CP appear to encompass the results of the proposed method across these three metrics.
5. The authors clarified that that standard CP controls error frequency but cannot bound costs. They also wrote that cost-aware CP averages out rare, high-cost events. The proposed framework, however, is able to bound tail risks.
6. The authors clarified that the MIMIC-IV dataset contains over 3,000 distinct class.

Concerns 1, 5, and 6 remain only partially resolved.

- Concern 1: As the paper stands it is unclear whether the automated cost matrix generation is meant to be a core contribution or not. If yes, then more justification and contextualization is necessary. If not, it should be de-emphasized (currently it is the second subsection in the methods section, after problem formulation) and positioned as a method for generating costs in order to conduct experiments.
- Concern 5: Although standard CP and cost-aware CP have shortcomings, there exist previous approaches such as Learn-then-test [1] that enforce multiple criteria, as well as approaches handling tail risks, e.g. [2]. The relationship between the proposed method and these previous works remains unclear.
- Concern 6: Despite the number of classes present in MIMIC-IV, reviewer kCEd’s point still stands about the limited evidence for generality of the proposed method. This would be strengthened by including results on additional domains.

[1] Anastasios N. Angelopoulos. Stephen Bates. Emmanuel J. Candès. Michael I. Jordan. Lihua Lei. "Learn then test: Calibrating predictive algorithms to achieve risk control." Ann. Appl. Stat. 19 (2) 1641 - 1662, June 2025.

[2] Chen, C., Shen, J., Deng, Z., & Lei, L. (2025). Conformal tail risk control for large language model alignment. Forty-Second International Conference on Machine Learning.

**Reviewer Scores:**

- SzPZ is very likely to keep their score. Their initial rating was positive. During the rebuttal, the authors provided clarification on the cost matrix generation and discussed properties of the score function.
- ALUi is about as likely as not to increase their score. The authors clarified that specifying costs is a prerequisite to high stakes decision-making. They also assured the reviewer that the comparison across is fair and added results using CC-CP. It is about equally likely that ALUi would increase their score (due to additional experimental results) as they would keep their score (due to remaining concerns about automatic cost generation).
- erEt is very likely to keep their score. The original review was already positive. The authors discussed the need to specify costs in order to perform high-stakes decision-making and argued that a comparison to Bayesian approaches is out of scope.
- kCEd is unlikely to increase their score as concerns about theoretical contributions and empirical evaluation still remain.

---

### Decision · Program_Chairs · 2026-01-26

Reject